



# Measurement of alkyl and multifunctional organic nitrates
# by Proton Transfer Reaction Mass Spectrometry
**Marius Duncianu[1], Marc David[1], Sakthivel Kartigueyane[1], Manuela Cirtog[1],**
**Jean-Francois Doussin[1], and Benedicte Picquet-Varrault[1]**
[1] Laboratoire Interuniversitaire des Systèmes Atmosphériques (LISA), UMR-CNRS
7583, Université Paris-Est-Créteil (UPEC) et Université Paris Diderot (UPD), France,
Correspondence to: benedicte.picquet-varrault@lisa.u-pec.fr
**Highlights**
• PTR-MS technique is proposed as a reliable measurement tool of individual organic
nitrates.
• Different clases of organic nitrates are characterized from the mass spectrometric point
of view.
• Different ionization modes and reagent ions are proposed for each type of organic
nitrates.
**Abstract**
A commercial PTR-ToF-MS has been optimized in order to allow the measurement of
individual organic nitrates in the atmosphere. This has been accomplished by shifting the
distribution between different ionizing analytes, $H_3O^+/H_3O^+(H_2O)_n$ or $NO^+/NO_2^+$. The
proposed approach has been proved to be appropriate for the on-line detection of individual
alkyl nitrates and functionalized nitrates. It has been shown that hydroxyl- and keto-nitrates
have a high affinity towards $NO^+$, leading to the formation of an adduct that allows to easily
identify the organic nitrate (R) with the $R-NO^+$ ion signal. The recorded sensitivities for both
ionization modes correspond to detection limits of tens of ppt min$^{-1}$ in the case of hydroxy-
and keto-nitrates. Alkyl nitrates exhibit a moderate affinity towards $NO^+$ ionization leading to
detection units of few hundreds of ppt and the highest sensitivity in $H_3O^+$ mode was obtained
for the water adducts signals. This method exhibits however lower capabilities for the
detection of PANs with detection limits in the ppb range.





## 1  Introduction

Organic nitrates are important species of the reactive nitrogen (NOy) budget in the
troposphere. They are formed in NOx rich air by the degradation of hydrocarbons initiated by
OH (daytime) and NO₃ (nighttime) radicals. Since organic nitrates have lifetimes of several
days or weeks (Perring et al., 2013), they can act as reservoirs for reactive nitrogen by
undergoing long-range transport in the free troposphere before decomposing and releasing
NOx. They play therefore a key role in the ozone formation as they sequester reactive
nitrogen in rich NOx regions and release it in regions where production of ozone may be NOx
limited. The significant impact of organic nitrates chemistry on ozone budget has been
confirmed by recent modelling studies (Curci et al., 2009; Horowitz et al., 2007).
In addition, several field studies have shown that both polluted and remote atmospheres
contain a large variety of organic nitrates which significantly affects the NOy budget (for
example, 35 - 40 % as reported by Buhr et al. (1990) and up to 70% according to a modeling
study performed by Madronich and Calvert (1990)). They are monofunctional alkyl nitrates,
PANs but also multifunctional alkyl nitrates (Browne et al., 2013; Fischer et al., 2000; Kastler
and Ballschmiter, 1999; Muthuramu et al., 1993; O'Brien et al., 1995). These latter include i)
hydroxynitrates which are formed by the oxidation of alkenes initiated by OH radicals and by
isomerisation processes of alkoxy radicals (Arey et al., 2001), ii) carbonyl-nitrates which are
produced by the NO₃-oxidation of alkenes but are also second-generation oxidation products
of hydrocarbons and iii) dinitrates which are also expected to be second-generation oxidation
products (Atkinson, 2000; Barnes et al., 1990; Roberts, 1990).
However, measurements of organic nitrates during field campaigns remain rare preventing a
precise evaluation of their impact on the NOy budget in a wide variety of environments. The
main problems in analyzing organic nitrates in the atmosphere are first the great complexity
of the mixtures due to the huge number of precursors and formation pathways and then the
fact that organic nitrates are explosive compounds and only few of them are commercial. So
standards have to be synthesized. These analytical difficulties also affect our capability to
study the chemical processes in which organic nitrates are involved during lab experiments
(e.g. in simulation chambers).



Table 1 summarizes, without the intent of being exhaustive, some keystone articles
concerning the organic nitrates analysis in field and laboratory studies.
Two approaches have been applied for the analysis of complex mixtures of organic nitrates
during field and lab experiments: first, analyses at the molecular scale which is a powerful
approach to elucidate mechanisms but has the drawback to be often limited to a low number
of species and then, functional group analyses which are very useful to assess global budgets
but bring poor information for the understanding of processes. The second approach allows to
shortcut the complexity of the organic nitrates chemistry governing their production,
transformation, and removal processes and assess directly the sum of peroxy nitrates ΣAN and
alkyl nitrates ΣAN (Day et al., 2002; Perring et al., 2010). The thermal dissociation (TD)
properties of different classes of nitrates was used as an analytical tool able to sketch the
global chemistry of $RONO_2$, same as the Laser-Induced Florescence (LIF), cavity ring down
spectroscopy (CRDS)(Paul et al., 2009), or cavity attenuated phase shift (CAPS)(Sadanaga et
al., 2016), were used to quantify the $NO_2$ issued from the organic nitrates decomposition. In a
similar way, the infrared spectroscopy (IR) was used to monitor the time dependent loss of
organic nitrates and measure the rate of homolytic O−N bond cleavage (Francisco and
Krylowski, 2005) employing specific absorption bands in IR (1638 cm$^{-1}$). Although
notoriously powerful analytical techniques, the LIF, CRDS, CAPS and the IR exhibit poor
capabilities in the individual quantification of organic nitrates mixtures due to their intrinsic
conceptual operation mode.
Historically, measurements of individual organic nitrates have been conducted by gas
chromatography coupled to electron capture detection - GC/ECD, both coupled or not (Atlas,
1988; Blake et al., 1999; Flocke et al., 2005; Fukui and Doskey, 1998; Muthuramu et al.,
1993), to a pyrolysis/luminol chemiluminescence - CL detector (Buhr et al., 1990; Fischer et
al., 2000; Flocke et al., 1991; Gaffney et al., 1999; Hao et al., 1994; Winer et al., 1974), to
electron impact mass spectrometry EI-MS (Luxenhofer and Ballschmiter, 1994; Luxenhofer
et al., 1994) or to negative ion chemical ionization mass spectrometry – NI/CI-MS (Beaver et
al., 2012; Tanimoto et al., 1999) using thermal electrons (e$^-$$_{th}$). Abundant in marine
environments, the halocarbons are highly sensitive towards the ECD detection (Fischer et al.,
2002; Fischer et al., 2000) and  may generate artifacts in the organic nitrates identification and
quantification (Fukui and Doskey, 1998).





Although the chromatographic separation represents an effective analytical tool, the organic nitrates identification relies mainly on the retention times, while the dedicated detectors are only able to confirm, in best case, the presence of the nitrate functional group in the molecule. Illustrative examples are given by the studies of Luxenhofer et al. (1994) and Kastler and Ballschmiter (1999) who succeed to analyze complex mixtures of alkyl and multifunctional organic nitrates by combining separation with liquid and gas chromatography and detection using the intense mass-to-charge (m/z) 46 fragment ion that corresponds to $NO_2^+$. The same study highlights possible interferences with dinitrophenols, nitro- and dinitrocresols, pentachloro-nitrobenzene and to a smaller extent nitrophenols.

Besides the poor temporal resolution, another major drawback of this method is represented by the recovery factor decline with longer times in the chromatographic columns. According to Roberts et al. (2002), the sensitivity of this method for PANs is characterized by a diminishing response factor proportional to the compounds retention time through the column. Measurement of functionalized (oxygenated) nitrates appears to be a greater challenge as the lower vapor pressures and stronger surface interactions of these molecules make sampling and chromatographic techniques less appropriate. Additionally, the detection of functionalized nitrates by electron impact mass spectrometry has proven to be difficult, mainly due to the instability and thus, the fragmentation, of the molecular ion formed (Mills et al., 2016; Roberts, 1990; Rollins et al., 2010).

Lately, the newly developed capabilities in Atmospheric Pressure / Chemical Ionization Mass Spectrometry (AP-CIMS; CIMS) (Huey, 2007; Perraud et al., 2010; Slusher et al., 2004; Teng et al., 2014) prone as potentially powerful tools in organic nitrates analysis. Several types of CIMS have been highlighted by the literature, the technique being currently in progress.

The AP-CIMS uses methanol as a proton source in order to generate $RH^+$ peaks of the parent ions. Therefore the nitrogen-containing ions are characterized by even m/z ratios, while the analytes containing only C, H, and O appear at odd m/z. Several PANs were identified as gas phase products of the α-pinene + $NO_3$ reaction with this technique (Perraud et al., 2010). The protonated molecular ions were identified and the most intense fragments in the MS/MS scan corresponds to losses of $NO_2$, $HNO_3$ and to a smaller extent $HOONO_2$. Hydroxynitrates and keto-nitrates have also been detected with this technique. For hydroxynitrates, $RH^+$ peaks as well as fragments corresponding to the losses of $H_2O$ and $NO_2 + H_2O$ in the MS/MS mode were detected (Perraud et al., 2010; Schoon et al., 2007; Tuazon et al., 1999).





The thermal dissociation–chemical ionization mass spectrometry (TD-CIMS) technique has
been used for measurement of PANs and other multifunctional organic nitrates by the means
of $I^-$ reaction (Lee et al., 2014; Slusher et al., 2004; Xiong et al., 2015). The obtained
carboxylate ion is unique for each parent species, and the only significant interference that has
been identified is at ppb levels of NO (Slusher et al., 2004). The $CF_3O^-$ was equally
successfully tested as ionizing source in a CIMS approach in order to identify hydroxy-
nitrates formed during the OH oxidation of alkenes in the presence of $O_2$ and NO (Bates et al.,
2014; Teng et al., 2014). The quantification of the nitrates formed was assured by a
complementary TD-LIF technique after subsequent GC separation. Other polyfunctional
organic nitrates were scarcely detected using this technique.
The Proton-Transfer Reaction Mass Spectrometry (PTR-MS) can be positioned as a subset of
CI. Its use in atmospheric research has expanded rapidly these last years but few studies have
tested this technique for the detection of organic nitrates (Aoki et al., 2007; Hansel and
Wisthaler, 2000; Inomata et al., 2013).
The recent study of Müller et al. (2012) considers that the quantification of PAN by PTR-MS
is difficult due to fragmentation. Less promising results are also reported by Aoki et al. (2007)
concerning the PTR ionization of $C_1$–$C_5$ alkyl nitrates. Considerable fragmentation occurs,
even at low field density (E/N) ratio (100 Td; $E$ being the electric field strength and $N$ the gas
number density; $1\,Td = 10^{-17}\,V\,cm^2$), the signal intensities of protonated alkyl nitrates,
$(ROH \cdot NO_2)^+$, being in best case, a few percent of those of the total ion signals. An increase of
the E/N ratio increases furthermore the fragmentation, providing common ions such as $NO_2^+$
and more characteristic ones such as $RO^+$. Other low intensity signals corresponding to $R^+$ or
$ROH \cdot H^+$ ions have been also recorded and could result from reactions of alkyl nitrates with
$H_3O^+(H_2O)_n$ clusters.
Hansel and Wisthaler (2000) have measured several PANs (PAN, MPAN and PPN) with
PTR-MS and have reported low detection limit (70 pptv, 15s integration time) and an overall
accuracy of 15 %. Worth noticing that the reported analytical sensitivities (15-25 counts $ppb^{-1}$
$^1$) have not been calculated for the molecular ion signal but for typical molecular fragments in
the case of the alkyl nitrates or signals of speculative subsequent reaction products of the
protonated molecular ions for PANs. These results are illustrative for the organic nitrates
overall affinity towards fragmentation due to chemical ionization in a low pressure drift tube





under high voltage. To our knowledge, polyfunctional organic nitrates have never been
detected using this technique.
So, it appears useful to better explore the possibility of detecting organic nitrates in real-time
with PTR-MS. In the present work, the detection of alkyl and multifunctional organic nitrates
(PANs, ketonitrates, hydroxynitrates) by this technique was investigated. For this purpose,
different organic nitrates were synthesized and mixtures at the ppb-ppm level were generated
in a smog chamber. A commercial PTR-ToF-MS instrument was used but the operating mode
as well as the chemical ionization reagent have been modified and optimized for each type of
organic nitrate in order to gain sensibility in the detection and to reduce fragmentation of ions.
Mass spectra have been carefully interpreted and detection limits have been also determined.

## 2    Material and methods

### 2.1    Instrumentation

A commercially available, high mass resolution PTR-ToF-MS instrument (Kore Technology -
'Series II' High Performance PTR-TOF-MS) was used in the present study. The mass
resolution of this instrument is >5000 M/δM. The PTR-MS is equipped with the newly
designed radio frequency ion funnel (RF) which is used to focus ions in the drift tube and
hence to increase the detection sensitivity (Barber et al., 2012). This ion funnel uses a series
of electrodes with progressively reducing aperture sizes. In addition to the standard dc
electrical field, an ac electric field is provided at a radio frequency creating a strongly
repulsive effective potential near the surface of the electrodes, which combined to the
reducing aperture sizes, serves to focus the ions radially. This system has been shown to
increase the sensitivity by one or two orders of magnitude (Barber et al., 2012).Worth notice
that operating in the RF mode modifies the dynamic range over which the drift tube is
operated and the contribution of the radiofrequency to the global effective E/N ratio remains
difficult to estimate (Barber et al., 2012).
The PTR-MS sampling line was designed in order to assure the highest transfer efficiency of
the compounds. This line is made of a 1.5 meter long Silcosteel® coated stainless steel
tubing, which has been shown to be appropriated for the transfer of low-level polar organic
compounds (Smith, 2003). This line was heated at 40°C in order to prevent adsorption of





compounds in the line. With a sampling flow rate of ~3 cm$^3$ s$^{-1}$, the residence time in the line
has been estimated to be lower than 2s. The PTR-MS inlet system is equally made with
Silcosteel® coated stainless steel tubing and is equipped with a temperature control unit
allowing heating up to 150°C.
## 2.2   Experimental strategy
In PTR-MS, proton transfer reactions with hydronium ions (H$_3$O$^+$) are typically used to ionize
compounds having a proton affinity (PA) higher than the one of water (691 kJ mol$^{-1}$, Hunter
and Lias, 1998):
$$H_3O^+ + R \rightarrow RH^+ + H_2O \qquad\qquad (R1)$$
In this case, the proton transfer reaction is exothermic and proceeds at a reaction rate close to
the collision rate (10$^{-9}$ cm$^3$ molecule$^{-1}$ s$^{-1}$; Flocke et al. (1991)). In order to avoid a massive
fragmentation of the analyte and consequently a decrease of the intensity and the specificity
of the analytical signal, the energy transferred during the reaction should be lower than the
bond energies of the compound of interest.
Only few available data can be found concerning the PA of organic nitrates which were
calculated using *ab initio* quantum mechanical methods exclusively. The PA of organic
nitrates have been estimated to: 740 kJ mol$^{-1}$ for methyl nitrate (Lee and Rice, 1992), 753 kJ
mol$^{-1}$ for ethyl nitrate (Kriemler and Buttrill, 1970), 748 kJ mol$^{-1}$ for methylperoxy nitrate
(Ravelo and Francisco, 2007) and 759–773 kJ mol$^{-1}$ for peroxyacetyl nitrate (Tureček, 2000).
These studies show that the PA of organic nitrates are higher than the one of water and so that
proton transfer reaction may occur. Furthermore, the functionalized nitrates concerned by the
current study are equally expected to easily protonate as it was shown that the presence of an
additional oxygen atom in the molecule enhances the overall PA (Ravelo and Francisco,

208   2007).

It is well known that other chemical processes can also occur in the PTR reactor (Blake et al.,
2009; de Gouw and Warneke, 2007). The H$_3$O$^+$ and R$^+$ can cluster with water molecules in
the sampled air. The water cluster ions formation in the PTR cell is generally considered as
problematic since their presence complicates the mass spectra treatment. Typically, their
formation is limited by high kinetic energy of the ions into the drift tube at E/N ratios superior
to 100 Td. However, the proton-transfer reaction of H$_3$O$^+$(H$_2$O)$_n$ clusters is more selective





than the $H_3O^+$ due to a higher PA (808 kJ mol⁻¹, Goebbert and Wentold (2004)). This property
may be valuable in our case as it allows a softer ionization of analytes. Recently Jacobs et al.
(2014) have used $H_3O^+(H_2O)_n$ ions (distribution centered around n=4) in PTR-CIMS
technique with the objective of monitoring isoprene derived hydroxynitrates and their
oxidation products. In order to evaluate the efficiency of the protonation mode by
$H_3O^+(H_2O)_n$ for the detection of other organic nitrates, a wide range of E/N ratios will be
tested.
The $NO^+$ chemical ionization has also been tested for the detection of organic nitrates during
our experiments. It has been shown to be a complementary, sensitive and reliable method for
the detection of some organic compounds such as ketones, aldehydes and alkenes by CI-MS
and SIFT-MS techniques (Knighton et al., 2009; Mochalski et al., 2014; Perraud et al., 2010;
Wang et al., 2004). $NO^+$ ions are produced from the ionization of dry air within the hollow
cathode discharge ion source. This process produces large amounts of $NO^+$ with only minor
amounts of $NO_2^+$, $O_2^+$, and $H_3O^+(H_2O)_n$ impurities (Knighton et al., 2009). The intensities of
$O_2^+$, and $H_3O^+(H_2O)_n$ are quite low, less than 1% of the primary $NO^+$ signal. In addition, it has
been shown that the intensity of the $NO_2^+$ impurity is controllable and dependant on how the
hollow cathode ion source is operated (Knighton et al., 2009). The ion source extraction
voltage and the hollow cathode discharge current are the two most important parameters.
Previous studies interrelated the PA of water to the $NO^+$ chemical ionization binding energies
(BE), creating an absolute $NO^+$ affinity scale (Cacace et al., 1997). The study used the $H_2O$-
$NO^+$ binding energy as a reference anchor between the two conjugated parameters and
concerned various classes of ligands (alkyl halides, alkyl nitrates, alcohols, nitro-alkanes,
nitriles, aldehydes, ketones, aromatic and heterocyclic compounds). With the exception of the
aromatic compounds, for which the $NO^+$ σ type complex structures are replaced by the π
complexes, the overall data exhibit a highly correlated linear dependence ($BE_{NO+} = 0.367$ PA
– 174.5 kJ mol⁻¹). When compared with protonation energies, the binding energies
characterizing the $NO^+$ chemical ionization are significantly lower (Cacace et al., 1997).
This characteristic offers the opportunity to generate soft ionizations with little fragmentation
of the product ions, which is of high interest for the present study. Reactions which may occur
are:

245                $NO^+ + R \rightarrow R^+ + NO$                          (R2a)



$\qquad$ $NO^+ + R \rightarrow R \cdot NO^+$ $\qquad$ (R2b)
$\qquad$ $NO^+ + RX \rightarrow R^+ + XNO$ $\quad$ (X=-H, -OH, -CH₃, -OR) $\quad$ (R2c)
As a function of the chemical affinity, $NO^+$ chemical ionization may follow several pathways.
The charge transfer reaction (2a) seems to be the most common process for compounds
having ionization energies slightly lower or close to that of NO (IE = 9.26 eV) (e.g. isoprene
(Karl et al., 2012) butadiene, benzene (Knighton et al., 2009), monoterpenes, terpenoids
(Amadei and Ross, 2011; Rimetz-Planchon et al., 2011) and phenols (Wang et al., 2004)).
This hypothesis is confirmed by Smith et al. (2003) who reported for a series of ketones (C₃-
C₁₁) that the yields of parent radical cation increase as the IE decrease.
The adduct formation pathway (2b) has been previously reported for the detection of C₃-C₄
alkanes, alkenes and terpenes (Diskin et al., 2002; Španěl and Smith, 1998), ketones (Smith et
al., 2003; Wang et al., 2004) and flavoring esters (Iachetta et al., 2010) by selected ion flow
tube mass spectrometry. The speculated mechanism which has been proposed to explain the
formation of the adduct involves the formation of an excited intermediate complex $(R \cdot NO^+)^*$
which is then stabilized by collision with the carrier gas (Smith et al., 2003). Evidently the
process is enhanced at upper PTR reactor pressure involving higher collision rates.
A third mechanism, similar to the protonation, involves the hydride, hydroxide, methyl or
alkoxy abstraction (2c), as reported for aldehydes, ethers, alcohols (Smith and Španěl, 2005)
terpenoids (Amadei and Ross, 2011) or several unsaturated alcohols (Karl et al., 2012;
Schoon et al., 2007; Wang et al., 2004).
The $O_2^+$ ionization can generate artefacts, particularly when the PTR-MS is operated in dry air
mode, as it proceeds via a dissociative charge transfer reaction producing parent cations $M^+$
but also fragment ions. In consequence, this process presents low interest for chemical
ionization of large molecules.
To summarize, several parameters will be tested in order to establish the optimal conditions
for the detection of each class of organic nitrates.
A thoughtful procedure was performed in order to establish the optimal measurement
conditions for each class of organic nitrates:
$\quad$ - $\quad$ First, the instrument has been used in a dual mode, employing alternatively $H_3O^+$ and
$\qquad$ $NO^+$ as a reagent ion, by shifting the GD source gas from water vapor to dry air. Dry
$\qquad$ air was supplied to the hollow cathode ion source using the alternative GD ionization





line. A third operation mode which consists in suppressing the GD source gas once the
plasma is established has also been tested. This last operational mode implies hence
the retro-diffusion of the sampling line dry air into the GD cavity and will be further
referenced as the Retro-diffusion mode (RDiff).
-    The ions intensity and distribution are sensitive to the extraction voltage used to inject
the reagent ions into the drift tube and to the hollow cathode discharge current. The
PTR-MS instrument was thus operated in a wide range of E/N (electric field to number
density of the gas) ratios (30 – 180 Td), hence shifting the distribution between the
different ionizing analytes in the protonation mode ($H_3O^+$/ $H_3O^+(H_2O)_n$) or into the dry
air mode ($NO^+$/$NO_2^+$).
-    The sampling line and the inlet system along with the drift tube was slightly heated
(40°C) and the reactor pressure ranged from 0.7 to 1.6 mbar. The sampling flow rate
(2.9 to 3.5 $cm^3$ $s^{-1}$) affects through the mass of the sampled analyte or through the third
body collision processes the instrument response. An enhanced influence of this
parameter may be noticed while operating in the RF mode.
-    The influence of radio frequency ion funnel mode on the detection of the various
organic nitrates has been tested by performing experiments with the RF mode on and
off.
Ideally, little or no fragmentation occurs in the case of a soft ionization process. The large
differences in ionization energy of the colliding species enhance the fragmentation of the
organic molecule. Therefore the ionizing gas species should correspond, especially for labile
molecules, to the analyte of interest in order to achieve the high yield ionizations with low
excess energy.
In addition, the soft ionization processes demand the usage of uncommon settings of the PTR-
MS device. The various effects of the altered PTR-MS tuning are termed in detail elsewhere
(Hewitt et al., 2003; Knighton et al., 2009). Supplementary information relating the ion source
discharge current and the extraction voltage influence are discussed in the Results section.
All reported signal intensities took systematically into account the background values,
representing in most of the cases negligible low values of a few counts per minute.
Background measurements are achieved before every single experiment by sampling the dry
synthetic air of the reaction chamber.





### 2.3 Normalizing analytical signal for $H_3O^+$ and $NO^+$ ionization
In this study, calibrated response factors of PTR-MS are determined for several types of
organic nitrates in both $H_3O^+$ and $NO^+$ ionization modes. For that purpose, known amounts of
organic nitrates are introduced in LISA simulation chamber (see section below). To quantify
the PTR-ToF-MS response factors (S; ncpm ppbv$^{-1}$), the ion signal ($I_{R+}$; ncpm) is divided by
the known concentration of organic nitrate (ppbv$^{-1}$), as measured by the FTIR in-situ
technique. Corrections of the product ion signal ($I_{R+}$) due to changes of the operational
conditions have to be taken into account, as shown by previous studies (Knighton et al.,

316 2009):

$$I_{R^+} = \left(\frac{i_{R^+}}{f_{IA^+}}\right) \left(\frac{P_0}{P}\right)^2 \left(\frac{T}{T_0}\right)^2 \qquad \text{(Eq. 1)}$$

The intensity of the product ion raw signal ($i_{R+}$) expressed in counts per minute is adjusted to
the variabilities of the ionizing analyte ($i_{IA^+}$) normalized to its averaged values ($\langle i_{IA^+}\rangle$),
($f_{IA^+} = i_{IA^+}/\langle i_{IA^+}\rangle$; $IA^+ = H_3O^+$ or $NO^+$) as it has been shown that both signals ($i_{R+}$ and $i_{IA+}$)
are related. The drift tube temperature (T) and pressure (P) are also included in this expression
to account for the small changes in the reaction time and gas number density that can occur
during measurements. In our experiments, the temperature was very stable (± 1 K) but the
PTR pressure was affected by the pressure in the simulation chamber which usually slowly
decreases because of the sampling from in the constant volume of the rigid chamber.
Due to the high abundance of the ionizing species, the direct measurement of $H_3O^+$ and $NO^+$
reagents was not possible as the ion counting signals (m/z 19 and 30) were regularly
saturated. The natural isotopic abundance properties of $^{17}O$ (0.038% of O), $^{18}O$ (0.200% of O)
and $^{15}N$ (0.368% of N) are used to overcome this drawback and evaluate the primary $H_3O^+$
and $NO^+$ ion intensity using the m/z 21 ($H_3^{18}O$) and 31 respectively ($\Sigma$ $^{15}NO^+$ and $N^{17}O^+$) (De
Laeter et al., 2003). The $I_{H30+} = I_{m/z21} \times 500$ and $I_{NO+} = I_{m/z31} \times 247$ formulas were applied as a
result of isotopic abundance probability calculations.
### 2.4 LISA Atmospheric Simulation Chamber.
Gaseous mixtures of organic nitrates at the ppb-ppm level were generated in the simulation
chamber at LISA. This chamber comprises a Pyrex reactor of 977 L equipped with a multiple
reflection optical system interfaced to a FT-IR spectrometer (Vertex 80 from Bruker). Details
of this smog chamber are given elsewhere (Doussin et al, 1997).





All experiments were conducted in the dark at 298 ± 2 K and atmospheric pressure. Mixtures
of organic nitrates were generated in synthetic air ($N_2$ 80% + $O_2$ 20%) by introducing a
known amount of the organic nitrate into the chamber and cross-monitored by long path *in*
*situ* FTIR and PTR-MS techniques. Concentrations of organic nitrates were checked from
their infrared spectral absorption bands. Integrated Band Intensities (IBIs; expressed in $cm^2$
molecule$^{-1}$ and calculated in decimal logarithm) and the spectral integration areas used to
quantify PAN like compounds and other nitrates are: $9.50 \times 10^{-18}$ for PANs (765–812 cm$^{-1}$),
$8.30 \times 10^{-18}$ for alkyl nitrates (820-900 cm$^{-1}$), $1.17 \times 10^{-17}$ for hydroxynitrates (816-882 cm$^{-1}$) ,
$1.08 \times 10^{-17}$ for ketonitrates (820-870 cm$^{-1}$). The IBIs values are given by the Beaver et al.
(2012) and Bates et al. (2014) studies for the PANs and the hydroxynitrates respectively,
whereas for the alkyl nitrates and ketonitrates the IBIs were calculated from repetitive
injections of known amount of analyte, in the current study. Worth notice the weak disparity
of cross sections concerning the characteristic -$ONO_2$ absorption band.
Before and after every single experiment, a cleaning procedure was applied in order to avoid
memory effects from an experiment to the next one. It consists in vacuum clean-up down to
$10^{-2}$ mbar for at least 10h and UV irradiation (340 and 360 nm fluorescent tubes) in order to
heat the reactor and improve desorption of semi and low volatile compounds from the walls.

## 356   2.5   Chemicals and gases

The simulation chamber was filled with dry synthetic air generated with $N_2$ (from liquid
nitrogen evaporation, >99.995% pure, <5ppm $H_2O$, Linde Gas) and $O_2$ (quality N45,
>99.995% pure, <5 ppm $H_2O$, Air Liquide). Supplementary cylinder air was used for the dry
air glow discharge generation in PTRMS (ALPHAGAZ™ 2; $H_2O$ < 0,5 ppm; $C_nH_m$ < 50 ppb;
$CO_2$ < 0,1 ppm; CO < 0,1 ppm; $NO_x$ < 10 ppb)
The alkyl nitrates considered in the present study (n-propyl nitrate - 97% Janssen Chimica;
AlkC3 and isobutyl nitrate - 96 % Sigma Aldrich; AlkiC4) were used as commercially
available, without further purification.
Ketonitrates were synthesized using Kames' method, (Kames et al., 1993): a liquid/gas phase
reaction in which the corresponding hydroxyketone is reacted with $NO_3$ radicals released
from the dissociation of $N_2O_5$ , at ice bath temperature, under dry condition. The carbonyl



nitrates and nitric acid were separated by multiple headspaces vacuum procedure. The
carbonyl nitrates' structure and purity were verified by FT-IR. The presence of $HNO_3$ is
regularly noticed as injected as by product of the carbonyl nitrates synthesis. 1-hydroxy-2-
propanone (95% Alfa Aesar) and 3-hydroxy-3-methyl-2-butanone (97% Sigma Aldrich) were
used for the synthesis of 1-nitroxy-2-propanone and 3-nitroxy-2-propanone respectively.
The hydroxynitrate was synthetized starting from the commercially available 3-bromo-1-
propanol (97% Sigma Aldrich). Its conversion to the analog iodide was performed by a
Finkelstein reaction with sodium iodide (Baughman et al., 2004) and the subsequent mild
conversion with $AgNO_3$ (Castedo et al., 1992) leads to the formation of 1-hydroxy-3-nitroxy-
propane (1OH3C3).
The PAN type compounds were generated in the simulation chamber from the gas-phase
oxidation of corresponding aldehydes by $NO_3$ radicals (Hanst, 1971). In our experiments,
peroxyacetyl nitrate (PAN) was formed from the oxidation of acetaldehyde (>99.5% Sigma
Aldrich). This reaction has been shown to produce mainly PAN with a yield close to 70%
(Doussin et al., 2003).
**3    Results and discussion**
**3.1    Influence of the operating conditions on the ionizing analytes signals**
As already termed, intensities and mixing ratios of the ionizing species are dependent on the
E/N ratio. For a standard drift tube, E/N is a well-defined quantity but when the instrument is
run in RF mode, the influence of the additional ac electric field on the E/N ratio has to be
taken into account and this is not obvious. Barber et al, 2012 have attempted to empirically
estimate an effective E/N by running the instrument with the RF mode on and off and by
seeking operating conditions for the RF mode on that match the performance obtained when
the RF mode is off. The criterion used to estimate the performance is the ratio
$[H_3O^+]/[H_3O^+(H_2O)]$. During our experiments, the additional ac electric field was fixed. Thus,
variations of E/N ratio result only from variations of the dc component of the drift tube.

Typically recorded distributions over a relevant range of E/N are illustrated in *Figure 1* for
both ionization modes, with and without the above described RF mode. When the RF mode is
not employed, the water cluster ion distribution obtained in this study (*Figure 1*) is in





reasonable agreement with those calculated and measured by de Gouw and Warneke (2007)
asserting that i) the $H_3O^+(H_2O)_2$ cluster is abundant for E/N ratios < 60Td, ii) the $H_3O^+(H_2O)$
signal is maximum around 80Td and iii) the $H_3O^+$ signal is predominant for values larger than
100Td. In the RF mode, it has been observed that the signal of $H_3O^+$ ion is much higher than
those of clusters for the entire range of E/N ratios. The $H_3O^+(H_2O)$ is one order of magnitude
lower than the $H_3O^+$ and one order of magnitude more abundant than the $H_3O^+(H_2O)_2$. As the
distributions of ions obtained with and without RF mode are quite different, it was not
possible to estimate an effective E/N as proposed by Barber et al., 2012. So E/N ratios have
been calculated without taking into account the RF contribution. In the case the RF mode is
on, E/N ratios are indicated with * (E/N*).
The $NO^+$ ionization mode has been found more promising in the RF mode, since the
abundance of the ionizing species is largely superior. Furthermore, the $NO^+/NO_2^+$ ratio
exhibits the highest values around 40Td in the RF mode, increasing the probability to form
the analyte-$NO^+$ adduct. Therefore, all further results presented in the current work in the $NO^+$
mode have been recorded with the RF mode and reported E/N* were calculated without
taking into account the ac radiofrequency contribution.

### 3.2  Alkyl-nitrates


### 3.2.1  $H_3O^+$ ionization mode


A series of tests has been conducted in order to seek the optimum operating conditions for the
measurement of alkyl nitrates with $H_3O^+$ ionization mode. In particular, the E/N ratio was
spanned in the 50-140 Td range. The intensity of the main signals obtained for n-propyl
nitrate (AlkC3) with the RF mode off has been plotted as a function of E/N in *Figure 2*. The
recorded mass spectra of AlkC3 are characterized by a high degree of fragmentation, even for
the lowest E/N (50 Td). The AlkC3 mass spectra recorded at the lowest extent of
fragmentation of the protonated molecular ion is illustrated by *Figure S1 in S.I.*. The same
result has been observed for AlkC4. Consequently, the sensitivity of the molecular ions
formed is low (see Table 2). The protonated alkyl nitrates are expected to adopt the ion-dipole
complex conformation ($ROH \cdot NO_2^+$ ; R1a) and not a covalently bound $RONO_2H^+$ one (Cacace
and de Petris, 2000).

427               $H_3O^+ + RONO_2 \rightarrow ROH \cdot NO_2^+ + H_2O$          (R1a)





The bound energy of these complexes was calculated by *ab initio* methods to be around 82 kJ
$mol^{-1}$ in the case of methyl nitrate (Lee and Rice, 1992).
The only other study (Aoki et al., 2007) describing a tentative of alkyl nitrate detection with
PTR-MS used E/N ratio spanning in between 96 and 147 Td. The low BE characterizing the
formed complex enables its decomposition in ROH and $NO_2^+$ by collision induced
dissociations. The recorded signals of $NO_2^+$ (m/z 46) and $RO^+$ (m/z 59 for AlkC3 and 73 for
AlkiC4) are probably formed by the mechanisms:
$$ROH \cdot NO_2^+ \rightarrow ROH + NO_2^+ \qquad (R3)$$
$$ROH \cdot NO_2^+ \rightarrow RO^+ + HONO \qquad (R4)$$
This statement is supported by the decrease of the protonated signal in conjunction with the
increase of the $NO_2^+$ with increasing E/N.
The presence of the m/z 43 (AlkC3) and respectively 57 (AlkiC4) signal was clearly
identified as the alkyl fragment ($R^+$) of nitrate by the means of high resolution mass
spectrometer allowing to differentiate between oxygen containing analytes and alkyl
fragments.
Due to the abundance of water clusters, higher in our study than typically, the R5 mechanism
is considered more likely, as proposed by Aoki et al. (2007). The formation of $R-OH.H^+$ ion
can also be explained by reaction R6 as proposed by Spanel and Smith (1997) This ion is then
expected to form the alkyl fragment ($R^+$) by reaction R7:
$$RONO_2 + H_3O^+(H_2O)_n \rightarrow R-OH \cdot H^+ + HNO_3 + nH_2O \qquad (R5)$$
$$RONO_2 \cdot H^+ + H_3O^+ \rightarrow R-OH \cdot H^+ + HNO_3 \qquad (R6)$$
$$R-OH \cdot H^+ \rightarrow R^+ + H_2O \qquad (R7)$$
A third option is to consider the direct formation of the alkyl fragment from the collision
induced dissociation of the protonated alkyl nitrates formed in reaction 1a into $R^+$ and $HNO_3$.
Further collisions of the alkyl fragment ($R^+ = C_3H_7^+$) might generate the loss of two
hydrogens and explain the weak signals at m/z 41 in the case of AlkC3.
$$C_3H_7^+ \rightarrow C_3H_5^+ + H_2 \qquad (R8)$$
It should be noticed that the analog process for the m/z 55 in the case of AlkiC4 cannot be
observed since it falls nearby a relatively abundant water cluster signal.



Results obtained with the RF mode are shown in Figure 3 for the AlkC3. The ionization
pattern of the analyte is slightly different from the one obtained without RF mode and
enables, adjacent to the formation of the protonated alkyl nitrate, the identification of other
specific signals like the adduct AlkC3·$H_3O^+$ formation. Characterized by a high degree of
fragmentation, the spectra contain the protonated ion-dipole complex ROH·$NO_2^+$ at m/z = 106
(R1a) with the highest sensibility above 60Td. The E/N ratio is calculated without taking into
account the RF contribution. The m/z 124 signal can be assigned to the AlkC3-$H_3O^+$ adduct
formation. An analog process is described in the literature for long alkanes (> C6) providing
M·$H_3O^+$ signals (Španěl and Smith, 1998), in contrast with the alkenes which are readily
protonated due their higher PA.
The alkyl ($R^+$) intermediary parent ion, described in R5 and R6 (R-OH·$H^+$) is equally
recorded, corresponding in the case of AlkC3 to the m/z 61 signal. The increase of the E/N
ratio over 60Td contributes to a higher degree of decay of this intermediate ion in the
particular conditions of the RF mode. The $RO^+$ ion at m/z 59 is equally present in the case of
AlkC3 confirming the existence of the R4 mechanism conducting to HONO formation.
To summarize, the use of $H_3O^+$ ionization is not really suitable for the detection of alkyl
nitrates as it leads to high fragmentation, even at low E/N ratios, hence generating intense
signals of common organic analytes, unsuitable for a reliable identification of these
compounds. In addition, if considering the more characteristic signals of the protonated ion-
dipole complex ROH·$NO_2^+$, the $H_3O^+$ ionization mode exhibits poor detection limit: 5 ppb
$min^{-1}$ for both AlkC3 (m/z 106) and AlkiC4 (m/z 120). This detection limit was estimated for
a mean signal/noise ratio of 3 at the chosen m/z.
In Table 2, are gathered the characteristic signals of each organic nitrate studied here as well
as their detection limits.

### 482 3.2.2 $NO^+$ ionization mode

The performances of the $NO^+$ ionization were also evaluated. The most promising results,
enabling the identification of characteristic signals of the studied alkyl nitrates, were obtained
in the RF mode. The scan of a large range of E/N* ratios confirmed that the highest sensibility
towards $NO^+$ induced ionization and identification is obtained around 40Td as illustrated in




Figure 3 for the AlkC3 case. As mentioned previously, the calculated E/N ratio does not take
into account the RF contribution to the ion funnel field.
The recorded mass spectra of the analyte of interest in terms of relative peak intensities as a
function of the m/z are represented by the thick black lines in Figure 4. It will be kept in mind
that the $NO^+$ and $NO_2^+$ signals illustrated above are the sum of the corresponding nitrate
fragments as well as the ions formed into the dry air plasma GD. The presence of $O_2^+$ (m/z =
32) and its interfering isotopic abundance signals at m/z = 33 and 34 has been observed and
could result in interfering signals.
For both alkyl nitrates, the adduct formation (2b) appears to be the main ionization
mechanism under these given conditions, leading to intense peaks at m/z = 135 (105+30) and
m/z = 149 (119+30) for AlkC3 and AlkC4 respectively (see Figure 4 and Figure S2). The
hydride abstraction $(R (-H))^+$ was also detected as a minor pathway (2c) at m/z = 104 and m/z
= 118  for AlkC3 and AlkC4 respectively. Despite a weak intensity, this signal can be used as
an interrelated identification signal for alkyl nitrates. The fraction of the abstraction channel is
at no time higher than 10% of the intensity of the adduct formation channel.
An intense peak corresponding to the alkyl fragment $(R^+)$ was observed in spectra at m/z 43
(AlkC3) and 57 (AlkiC4). The alkyl fragment signal is followed by a downward series of
signals at m/z 41 and 39 in the case of AlkC3 and at m/z 55 and 53 for the AlkiC4.
Considering the collisions of the alkyl fragment mechanism generating the loss of two
hydrogens (R8), as proposed by Aoki et al. (2007) for the protonation mode, could explain the
recorded signals.
In conclusion, the $NO^+$ ionization mode appears to be more suitable for the detection of alkyl
nitrates than the $H_3O^+$ ionization mode as it produces two characteristic signals,
corresponding to the adduct formation $(M.NO^+)$ and the hydride abstraction $(M(-H)^+)$. The
detection limit for the $M.NO^+$ signal was estimated to be 205 ppt $min^{-1}$ for AlkC3 (m/z 135)
and 126 ppt $min^{-1}$ in the case of AlkiC4 (m/z 149) (see Table 2). Although superior to the
protonation mode, the $NO^+$ ionization exhibits a poor sensitivity. For measurements in real
atmosphere where alkyl nitrates mixing ratios are usually in the ppt level, accumulations for
several hours will be necessary.



## 3.3 Hydroxy-nitrates

### 3.3.1 The $H_3O^+$ ionization

As for alkyl nitrates, the measurement of hydroxy-nitrates by PTR-MS has been tested with $H_3O^+$ ionization mode, with and without RF mode. For this purpose, 1-hydroxy-3-nitroxy-propane (1OH3C3) was synthesized and used as template. In the classical mode (RF off), the main signals which have been attributed to 1OH3C3 are: m/z 43 ($C_2H_3O^+$), m/z 104 ($M_{1OH3C3}(-OH)^+$), m/z 122 ($M_{1OH3C3}.H^+$), m/z 139 ($M_{1OH3C3}.H_2O)^+$). Intensities of these signals have been plotted as a function of E/N in *Figure 5*. As expected, the intensity of the signal m/z 43 increases with high E/N ratio while protonation and water adduct formation are favored by low E/N. A possible explanation for the formation of the ion $M_{1OH3C3}(-OH)^+$ is a loss of $H_2O$ after protonation of the hydroxyl-nitrate, as suggested by several studies for alcohols (Schoon et al., 2007; Smith et al., 2012; Tuazon et al., 1999).

When the RF mode is on, the same signals (43, 104, 122, 139) were observed but with different relative abundance. The influence of E/N on the intensity of these signals has been plotted in *Figure 6*. Above 40 Td, the signal characterizing the water adduct formation (m/z = 139) is more abundant in the RF mode with a maximum around 45Td.

The water clusters are probably abundant in the PTR reactor due to the low E/N ratios and dissociate only into the RF region after the ionization occurs.

The mass spectrum illustrated by Figure S3 was recorded in absence of the RF funnel and is dominated by fragments like m/z 59, 60, 73, while the specific signal of the protonated molecule m/z 122 is present close to the noise limit.

The above described water adduct formation (m/z = 139) is competed by a $H_3O^+$ adduct (m/z = 140), equally present into the spectra. The ionization is probably induced by the expected high levels of water cluster, following an analogous process with the one described in R5. This involves the release of $HNO_3$ and arise the signal at m/z 77 $(C_3H_6OH)-OH\cdot H^+$ and subsequently at m/z 59 with the loose of supplementary water (R7). Further collisions of the above mentioned ions may well explain the signals at m/z 75 and 57, generated by the loss of two hydrogens (R8).

In RF mode, the mass spectrum obtained for 1OH3C3 at E/N* =45 Td (corresponding to the operational condition for which the signal at m/z 139 is the most intense) was illustrated in





*Figure 7*. It is worth to notice that all the other competitive ionization processes are strongly
diminished in comparison to the spectrum obtained in classical mode. Only the specific signal
at m/z = 104 exhibit a stronger signature compared to the case above. The fragmentation
pattern is strongly diminished in this case.
To conclude, it has been observed that protonation of the hydroxynitrate is a minor process in
comparison to fragmentation and to adduct formation. The use of the RF mode significantly
reduces the fragmentation for the benefit of the $M.H_2O^+$ adduct formation. The lowest DL (80
ppb min$^{-1}$) was obtained at 45Td* in the RF mode for the signal corresponding to the water
adduct formation at m/z 139. The same signal in absence of the RF effect is at least twice
weaker in terms of sensibility. Sensitivities of other specific signals are equally proposed for
comparison in Table 2.

### 3.3.2  The NO$^+$ ionization

The detection of hydroxy-nitrates in NO$^+$ ionization mode has been evaluated for the first
time, by varying the E/N ratio (39 – 44 Td) and by studying the influence of the RF mode. As
stated before the highest sensibility towards the NO$^+$ adduct formation are obtained in the RF
mode, most likely due to the higher abundance of the ionizing species, as already seen in
Figure 1.
Results obtained with RF mode have shown in *Figure 8*. Main signals which have been
observed are: 151 and 167 which have been attributed to $M.NO^+$ and $M.NO_2^+$ adducts
formation and 43 ($C_2H_3O^+$) which is characteristic of fragmentation. Although weak, the
hydride abstraction (R2c) leading to a signal at m/z 120 has also been observed in the
recorded spectra. This process has already been observed in previous studies for various
saturated and unsaturated alcohols ($C_{1-10}$) and phenol (Karl et al., 2012; Schoon et al., 2007;
Spanel and Smith, 1997) using SIFT and PTR-MS techniques. The hydride ion transfer of
these compounds generates the corresponding carboxy ion (and HNO), while the hydroxide
ion transfer gives the corresponding hydrocarbon ion (and $HNO_2$). Assuming the
corresponding ionization of hydroxynitrate in the case of the 1OH3C3, would involve the
formation of the $(O_2NO-C_3H_6)-O^+$ at m/z 120 and $(O_2NO-C_3H_6)^+$ at m/z 104 but their spectral
signature is marginal into the spectra.





As expected, the intensity of the signal 43 increases with increasing E/N ratio. The opposite
tendency was observed for adducts. The mass spectrum obtained at E/N* = 41 which
corresponds to the highest sensitivity for the m/z 151 signal is shown in *Figure 9*.
Beside the above mentioned characteristic signals, other specific mechanisms could be
associated to the spectral signature of 1OH3C3. The mechanism is reviewed and suggested by
(Harrison, 1999) for the particular case of $NO^+$ ionization of primary alcohols. In this study,
an additional product $(R (-2H) + NO^+)$ has been observed which would correspond in our case
to an m/z = 149, hypothetically formed by the oxidation of the alcohol to the corresponding
aldehyde and its subsequent ionization. Although relatively weak the signal is systematically
recorded in the mass spectra of the hydroxy-nitrate. The 45 and 67 ions are equally present in
mass spectra and can be attributed to fragmentation.
The detection limit obtained in $NO^+$ ionization mode at E/N ratio of 41 Td* for the $M.NO^+$
signal at m/z 151 was quite low (37 ppt.min$^{-1}$) showing that this ionization mode is more
suitable than $H_3O^+$ mode for the measurement of hydroxy-nitrates. As shown in Table 2, the
hydrogen abstraction signal (m/z 120) is at least one order of magnitude less sensitive.

## 3.4    Keto-nitrates

### 3.4.1    The $H_3O^+$ ionization

Two distinct keto-nitrates were synthesized and characterized in the current study, 3-nitroxy-
2-propanone (KnC3) and the 3-nitroxy-3-methyl-2-butanone (KnC5).
In order to identify the optimal conditions for the detection of ketonitrates in $H_3O^+$ ionization
mode, tests were performed by varying E/N ratios into a large domain, from 45 to 170 Td
(*Figure 10*).
Several signals which can be attributed to keto-nitrates have been detected, as illustrated in
*Figure 10* for KnC3: At low E/N ratios, which correspond to the $H_3O^+(H_2O)_2$ controlled
regime, the formation of two keto-nitrate water cluster adducts, $M_{kn} \cdot H_3O^+$ (m/z=138) and
$M_{kn} \cdot H_3O^+(H_2O)$ (m/z=156) has been observed. Worth notice the reduction of these specific
signals with the decay of the preeminent ionizing analyte $H_3O^+(H_2O)_2$ (m/z= 55) signal. The
literature available data expect that for polar compoundsthe ionization process may follow



two similar pathways, by proton-transfer (R9a) and by ligand–switching reaction (R9b) (de
Gouw et al., 2003).
$$M_{kn} + H_3O^+(H_2O)_n \rightarrow M_{kn}H^+ + (n+1)\ H_2O \qquad (R9a)$$
$$M_{kn} + H_3O^+(H_2O)_n \rightarrow M_{kn}H^+(H_2O)_m + (n-m+1)\ H_2O \qquad (R9b)$$
The bound energy of the cluster ions formed in the reaction above (R9b) is weaker than the
water cluster bond. In the given conditions, there is a high probability that the drift tube
dissociative effect can split the formed cluster ions and lead to the formation of $MH^+$ and
$MH^+ \cdot (H_2O)$. Since the cluster ion distribution may be governed by the water vapor drift tube
loads, the detection efficiency can be humidity reliant in this particular case (de Gouw et al.,

614 2003).

Redistribution processes among the various precursor ions formed into the glow discharge can
equally occur, leading to the formation of auxiliary hydrated ions such as $NO^+(H_2O)_n$, able to
produce ligand-switching reactions alike the ones presented above (R9b).
For intermediate E/N ratios, where $H_3O^+(H_2O)$ water cluster is the key analyte, we notice that
the signal corresponding to $M_{kn} \cdot H^+$ (m/z=120) is maximum. In addition, two other signals
corresponding to $M_{kn} \cdot NO^+$ (m/z=149) and $M_{kn} \cdot NO_2^+$ (m/z=165) adducts formation have also
been observed and rise up to a maximum for these intermediate E/N ratios. These adducts can
be explained by the formation of $NO^+$ and $NO_2^+$ analytes in the GD, which increases with the
increasing E/N ratios.
For high E/N ratios, the m/z 43 ($C_2H_3O^+$) signal which is a common fragment of organic
compounds, strongly increases. This reveals that mainly fragmentation occurs, making this
region not suitable for the detection of ketonitrates.
From these results, the intermediate E/N ratios (70 - 80 Td) appear to be the most suitable for
the detection of keto-nitrates in $H_3O^+$ ionization mode. Mass spectra obtained at E/N=75Td
which corresponds to the highest sensibility of the $M_{kn} \cdot H^+$ signal are shown in *Figure 11* for
KnC3 and in S.I. Figure S4 for KnC5 respectively.
For KnC3, the most intense signal corresponds to $M_{kn} \cdot H^+$ (m/z=120). As discussed above,
additional processes occur which are responsible for other signals: $M_{kn} \cdot NO^+$ adduct formation
at m/z 149 and fragmentation at m/z 43 ($C_2H_3O^+$). These characteristic signals are tailed by
their corresponding isotopic abundance signals mainly due to $^{13}C$ isotope at m/z=121 and 150.



The already discussed reactions R5 and R6 could explain the intense signal of m/z 75 since
the resulting $(C_3H_5O)\text{-}OH\cdot H^+$ ion seems the accurate match of this signal.
Worth notice that due to the low E/N ratios, imposed by the breakability of organic nitrates, in
the above illustrated mass spectra examples, the intensities of the signals m/z 19 ($H_3O^+$) and
37 ($H_3O^+(H_2O)$) are of the same order of magnitude while the m/z 55 ($H_3O^+(H_2O)_2$) is one or
two orders of magnitude lower.
In the case of KnC5, the $M_{kn}\cdot H^+$ (m/z=148) signal is not the most intense one, suggesting that
fragmentation is more favored that for KnC3 under similar conditions. The characteristic
fragmentation pattern of this analyte exhibits a characteristic m/z 85 signal corresponding to
the $C_5H_9O^+$ group after the cleavage of the $NO_3$ fragment. An analogous process is described
by Aoki et al. (2007) asserting the alkyl group as the main signal in the mass spectra of alkyl
nitrates. The abundant m/z 59 fragment ($C_3H_7O^+$ or $C_2H_3O_2^+$) is most probably a result of
subsequent fragmentation/recombination processes.
The intense m/z=103 signal could be ascribed to the already mentioned R5 and/or R6, both
conducting to the formation of a $(C_5H_9O)\text{-}OH\cdot H^+$ ion with release of $HNO_3$ and water. Worth
notice that the elimination of supplementary $H_2O$ from the resulting complex could also
contribute to the m/z 85 signal ($C_5H_9O^+$).
In the RF mode, the water clusters distribution is dominated by the $H_3O^+$ ion, for the entire
range of E/N* ratios. Its signal is the most abundant, while all the other $H_3O^+(H_2O)_n$ species
are at least one order of magnitude lower. The redistribution of water cluster in the RF mode
modifies obviously the keto-nitrates' fragmentation pattern and the burden of each ionization
channel in a similar way as in the case of hydroxy-nitrates. No straightforward correlation can
be made in between the two modes, showing the complexity of the ion funnel influence.
The most noteworthy difference in the RF mode resides in the presence of an intense signal
corresponding to a $H_2O$ adduct formation (m/z 137 and 165 respectively), similar to the
hydroxy-nitrate, with a maximum intensity around 48 Td (Table 2) as calculated without
taking into account the RF contribution.
With the same PTR entry voltage, the RF mode activation induces the enhancement of the
fragmentation due to higher input energies and the maximum of the protonated ketonitrate
signal glides towards 48 Td. Making the assumption that the highest sensibility is related in
the two modes to an analogous E/N state, a coarse assessment of the RF mode contribution is



estimated (25 - 35 Td) as the difference between the two modes maximum sensibility for the
protonated molecular ion signal.
To conclude, it has been observed that protonation of ketonitrates is the main ionization
process in the absence of a RF field. The use of the RF mode modifies the fragmentation
patern and enhances the mechanism leading to the $M.H_2O^+$ adducts formation. The recorded
sensitivities listed in Table 2 for the $H_3O^+$ ionization mode, to a DL of 70 - 80 ppt min$^{-1}$ for
the protonated signal of KnC3 (m/z 120). A similar DL is reached by the water adduct signal
at m/z 137 under the influence of the RF ion funnel. The highest sensibility for the KnC5
identification is attained by the water adduct signal at m/z 165 recorded at 48Td in the RF
mode with a DL of 20 ppt min$^{-1}$. Sensitivities of other specific signals are equally proposed
for comparison in Table 2.

### 677    3.4.2  The NO$^+$ ionization

For the first time, the detection of keto-nitrates using NO$^+$ ionization was tested. In a similar
way as for the protonation process, mass spectra of KnC3 and KnC5 were recorded as a
function of E/N ratios. The previously proven affinity of NO$^+$ coupling -C(O) functional
group in ketones to form adducts (Smith et al., 2003) was for the first time tested in a PTR
reactor. The adduct formation was successfully achieved in the presence of NO$^+$ for both
studied ketonitrates, for which very similar IE are expected.
In the given ionization mode and considering the higher IE of ketonitrates, the adduct
formation is expected to prevail over the charge transfer, the literature stating that the yield of
parent radical cation formation seems to be anti-correlated with the IE of the ketones, the
lowest IE analytes, presenting the highest probability for the charge transfer reaction (Smith et
al., 2003).
Following the same approach as in the previous case, the E/N* region was set in order to
provide the highest $NO^+/NO_2^+$ ratio. The *Figure 12* is presenting the influence of the E/N ratio
variation in the RF mode over the analyte signal of interest. We notice that the NO$^+$ profile
sharply increasing together with the $NO^+/NO_2^+$ ratio is joint by the adduct signal.
We plot as an example, the KnC5 adduct signal at m/z=177 (147+30) as a function of two
fragments (m/z= 43 and 85) having potentially, as explained above, different formation
pathways. A decrease of the fragmentation yield in favor of the adduct formation in the PTR





reactor is observed over the 35 Td shoulder. The high sensitivity of the instrumental setup for
the identification of ketonitrates is quantified in Table 2.
Due to the rising incidence of the $NO^+$ and $NO_2^+$ ions, simultaneously formed in the GD, the
signals corresponding to the $M_{kn} \cdot NO^+$ and $M_{kn} \cdot NO_2^+$ adducts formation is equally
strengthened (m/z=149 and 165 for the KnC3 - *Figure 13* and m/z=177 and 193 for the KnC5
– see S.I. Figure S5 respectively). All these characteristic signals are tailed by their
corresponding isotopic abundance signals due mainly to $^{13}C$ isotope at m/z=121, 150, 166,
178 and 194 respectively.
The KnC3 spectrum was recorded at E/N* = 40Td as shown in *Figure 13*. We notice a low
fragmentation pattern of the analyte with a major adduct signal contribution at m/z= 149. The
existence of a secondary (water controlled) competitive process might be revealed by the
existence of the minor protonated signal at m/z 120 backed by the characteristic fragment at
m/z =75 as described by R5-7.
In an analog way, in *S.I. Figure S5,* the mass spectrum of KnC5 is plotted for the instrumental
setup corresponding at the lowest fragmentation (E/N = 36 Td). We notice the distinct signal
of the formed adduct (m/z =177) as well as, to a lower extent, the characteristic signals
described above (m/z= 59, 85, 103) which are most likely formed following the mechanisms
described in the previous paragraph, since the water is ubiquitously present in the system.
The best results related to the $NO^+$ ionization are related typically to the highest $NO^+/NO_2^+$
ratios in the RF mode as seen in Table 2. The $NO^+$ high affinity towards adduct formation is
confirmed by the low DL achieved for these highly characteristic signals, 30 and 40 ppt min$^{-1}$
for the adducts formed at m/z 149 and 177 respectively at an E/N* ratio < 45 Td.

### 3.5  PANs

The detection of PANs with PTR-MS has been tested by generating the peroxyacetyl nitrate
(PAN) from the well-known $NO_3$-oxidation of acetaldehyde in the simulation chamber. The
disadvantage of this procedure despite the 70% PAN formation yield (Doussin et al., 2003) is
that it generates several bi-products (nitric acid, formaldehyde…) leading to more complex
mass spectra. The PANs family analytes were all obtained by in-situ generation using the
corresponding aldehydes. In order to overcome any judgment error, the present study will





only illustrate the PAN mass spectra while for the other analogous formed PAN like products
(P2MCrN and P3MCrN) we will only discuss about the mass spectra signals which are
assigned to them.

### 729    3.5.1   The H$_3$O$^+$ ionization

The optimal conditions for the detection of PANs in H$_3$O$^+$ ionization mode were explored by
varying the E/N ratios from 55 to 120 Td. The most promising results were obtained in the
case of PAN around 85Td where the protonated signal of PAN was recorded as we can depict
from Figure S6 in S.I.
In an analogous way with the reaction R1a, the protonated peroxy-nitrates are equally
expected to form low energy gas phase ion-dipole complexes (ROOH·NO$_2^+$) with bound
energies of 92 kJ mol$^{-1}$ in the case of methyl-peroxynitrate (Ravelo and Francisco, 2007).
Previous studies (Hansel and Wisthaler, 2000) have proposed a speculative decomposition of
the protonated PAN:

739                 $RC(O)OONO_2H^+ + H_2O \rightarrow RC(O)OOHH^+ + HNO_3$          (R10)

The same mechanism is equally indicated by later studies (Aoki et al., 2007), giving the mass
spectral signal at m/z = 77, as recorded in the case of the PAN (Figure S6). According to their
statements the other protonated PANs parent compound generated in our study (C$_4$H$_7$-
C(O)OONO$_2$H$^+$, m/z = 162) should lead to representative ionic signals of type C$_4$H$_7$-
C(O)OOHH$^+$, m/z = 117, which are indeed present in the deconvoluted mass spectra of our
analytes. Moreover, the fact that this signal is absent during the NO$^+$ ionization mode
(characterized by inferior H$_2$O levels) may support this hypothetic decomposition of the
PANs.
Another likely mechanism which, unexpectedly, may lead to the same analytical signal is
reviewed by (Roberts, 1990) proposing the unimolecular decomposition of PANs to form the
corresponding C$_{n-1}$ alkyl nitrate. In our particular case the reaction may be written as:

751                 $C_5H_7-C(O)OONO_2 \rightarrow C_4H_7-ONO_2 + CO_2$          (R11)

Assuming the presence of both processes, the resulting alkyl nitrate may interfere with the
above mentioned m/z=117 signal in the case of a charge transfer reaction. More likely, in the
case of a protonation process, the obtained m/z=118 signal derived from the alkyl nitrate will
enhance the isobaric signal due to the $^{13}$C isotopic abundance of the analyte obtained by PAN





decomposition. The signal of m/z 118 seems indicative of the presence of a double peak
which might confirm the occurrence of both processes. This fact is equally confirmed by the
$I_{118}/I_{117}$ measured ratio (0.19 - 0.23 for the P2MCrN and 0.52 - 0.57 for the P3MCrN) which
is significantly different from the expected $^{13}C$ isotopic abundance for a $RC(O)OOHH^+$ type
compound (0.06). This difference suggests the presence of this kind of analytical interference,
more pronounced in the case of P3MCrN.
However the PAN decomposition path is considered to be several hundred times slower than
the bond homolysis channel:
$$C_4H_7\text{-}C(O)OONO_2 \rightarrow C_4H_7\text{-}C(O)OO^. + NO_2 \qquad (R12)$$
The recombination of $C_4H_7\text{-}C(O)OO^.$ (m/z = 115) with highly abundant $H^+(H_2O)$ and
$H^+(H_2O)_2$ could generate the signals at m/z 133 and 151 respectively. Additionally, the high
levels of $NO_2^+$ (m/z = 46) may favor the presence of $NO_2^+\cdot H_2O$ (m/z = 64) which can
potentially quench the $C_4H_7\text{-}C(O)OO^.$ radical to give an adduct signal at m/z = 179.
In order to certify the mono-nitrogen containing analytes for several ion signals in the
spectrum, an analysis similar to the one performed by Inomata et al. (2013) which consists in
subtraction of the isotopic effect of $^{13}C$ by calculating $I_{even}$ / $I_{odd}$ ratios of an ion signal at an
even m/z [M] to the adjacent ion signal at an odd m/z [M+1].
As described by Table 2 the normalized sensitivity for the protonated PANs is weak spanning
few counts $ppb^{-1}$ $min^{-1}$ for all three considered PANs.

### 3.5.2 The $NO^+$ ionization

Unlike the keto-nitrates, the PANs present a low sensibility towards detection in $NO^+$
ionization mode and generally a complex chemical equilibrium in classical PTR analysis.
In the $NO^+$ ionization mode, the signals corresponding to the adduct formation is barely
noticed for the entire range of E/N considered. However the presence of weak signals at m/z =
191 and 207 for both P2MCrN and P3MCrN, are indicative of the presence of PAN $NO^+$ and
PAN $NO_2^+$ adducts respectively.



## 4   Conclusions

Organic nitrates play a key role in atmospheric chemistry as they act as reactive nitrogen reservoir species. The use of PTR-MS for the measurement of VOCs has expanded a lot in atmospheric research these last years but few studies have investigated the performances of this instrument for the detection of organic nitrates. These studies have shown that this technique exhibits poor performances (high fragmentation and poor sensitivity) when it is run in classical mode. In the present work, the detection of alkyl and multifunctional organic nitrates (PANs, ketonitrates, hydroxynitrates) by this technique has been studied by operating the instrument in the classical mode ($H_3O^+$ as ionizing species) but also by testing an ionization induced by $NO^+$. This study has shown that a soft ionization by $NO^+$ using low E/N ratios (<50 Td) promotes the $R-NO^+$ adduct formation and minimizes the fragmentation, favoring the identification of molecular composition. In addition, the versatility of the instrument allows an easy change of the chemical ionization source, from $H_3O^+$ to $NO^+$ by simply replacing water vapor by dry air in the glow discharge. This is very useful for a double identification of the organic nitrates. In terms of sensitivity, the $NO^+$ adduct ionization mode appears to be the most sensitive for the detection of hydroxy- and ketonitrates with detection limits in the range of tens ppt/min. This sensitivity is suitable for measurement of organic nitrates during lab studies but also in ambient air. The detection of alkyl nitrates and PANs with PTR-MS is less sensitive, with detection limits in the range of hundreds ppt/min, whatever the ionization source used. For these two classes of nitrates, accumulations over longer periods will be necessary for measurements in ambient air.

A crucial aspect to be taken into account in further studies for lab and field measurements deployment of the method will be the effect of the humidity of the sampled air. In addition, longer acquisition time, elimination of interfering ionization paths by selective ionization sources and softer ionization sources could improve the technique's performances.

## 5   Acknowledgements

This work was supported by the French National Agency for Research (Project ONCEM-ANR-12-BS06-0017-01), by the European Community within the seventh Framework Program, section "Support for Research Infrastructure-Integrated Infrastructure Initiative":





EUROCHAMP-2 (RII3-CT-2009-228335) and by the Région Ile de France. The authors also
thank Fraiser Reich and Kore Company for their advices in the use of the PTR-MS.

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





*Table 1 Summarized analytical approaches into the organic nitrate analysis*

| Type of study | Analytes | DL / time or normalized sensitivity | Ionizing species | Experimental setup | Study |
|---|---|---|---|---|---|
| **Synthesis** | PAN | < 40 ppb | $O_3$+NO<br>KOH / $O_3$+NO | CL $NO_x$<br>IC KOH / CL $NO_x$ | Winer et al. (1974)<br>Grosjean and Harrison (1985) |
| **Field campaign** | PAN | 10-80 ppt / 1 min | luminol | CL $NO_2$ | Gaffney et al. (1999) |
| **Synthesis** | $C_3$–$C_5$ alkyl nitrates<br>$C_2$-$C_4$ hydroxy nitrates<br>$C_2$-$C_4$ dinitrates | | $e^-$ / luminol | GC-ECD –CL $NO_2$ | Hao et al. (1994) |
| **Laboratory / Field campaign** | | | | | |
| **Field campaign** | ΣAN, ΣPN | 30-90 ppt | - | TD-LIF | Day et al. (2002) |
| **Field campaign** | ΣAN, ΣPN | 30-90 ppt | - | TD-LIF | Perring et al. (2010) |
| **Field campaign** | $C_3$–$C_5$ alkyl nitrates | 10 ppt | $e^-$<br>$e^-_{th}$ | GC-ECD<br>GC-NICI-MS | Atlas (1988) |
| **Field campaign** | $C_2$–$C_3$ Alkyl nitrates<br>PAN, PPN, | 10 ppt | $e^-$ | GC-ECD | Fukui and Doskey (1998) |
| **Synthesis;**<br>**cis-2-butene+OH (NO)** | Alkyl nitrates<br>α, β -hydroxynitrates, dinitrates | - | $e^-$ | GC-ECD | Muthuramu et al. (1993) |
| **Peroxyalkyl +$NO_2$** | PAN, PPN, PiBN, MPAN, APAN, PnBN, PBzN | 2 ppt | $e^-$ | GC-ECD | Flocke et al. (2005) |
| **Field campaign** | $C_4$–$C_{14}$ alkyl nitrates<br>($C_2$-$C_8$)-alkyl- and phenyl-alkyl-nitrates | 1 ppt | $e^-$ | GC-ECD<br>GC-EI-MS | Luxenhofer and Ballschmiter (1994)<br>Luxenhofer et al. (1994) |



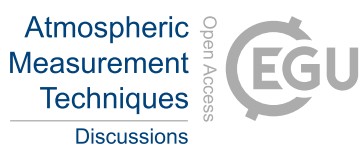

| | Compounds | Reagent ion | Technique | Sensitivity | Reference |
|---|---|---|---|---|---|
| **Field campaign** | $C_6$-$C_{13}$ alkyl nitrates | $e^-$ | HPLC / GC-ECD | | Fischer et al. (2000) |
| | $C_3$-$C_6$ dinitrates, $C_2$-$C_6$ hydroxynitrates | | GC-EI-MS | | Kastler and Ballschmiter (1999) |
| **Field campaign** | PAN, PPN, MPAN, | $e^-$ , $e_{th}$ | GC-ECD, GC-NICI-MS | 15 ppt / 10 min | Tanimoto et al. (1999) |
| **Synthesis; alcohols+$N_2O_5$** | $C_1$-$C_8$ alkyl nitrates, ketonitrates, hydroxynitrates, dinitrates | $e^-$ , $CH_5^+$ | GC-ECD, GC-$NO_y$, CI-MS | - | Kames et al. (1993) |
| **$NO_3$ + α-pinene** | Carbonyl hydroxi-nitrates or PANs | $H^+$ (MeOH) | APCI-MS | - | Perraud et al. (2010) |
| **Peroxyacetyl +$NO_2$** | PAN, PPN, MPAN | $I^-$ | TD-CIMS | 10 ppt /15s | Slusher et al. (2004); Huey (2007) |
| **$C_2$-$C_8$ alkenes + OH ($O_2$, NO)** | β-hydroxy-nitrates | $CF_3O^-$ | ToF CIMS, MS-MS CIMS | 19-50 counts/ppb | Teng et al. (2014) |
| **Field campaign** | PAN, PPN, MPAN, | $O_2^+$, $H_3O^+$, $(H_2O)_2H^+$, $H_3O^+$ | SIFDT*, PTRMS | 25 counts/ppb | Hansel and Wisthaler (2000) |
| **Synthesis** | $C_1$-$C_5$ alkyl nitrates | $H_3O^+$ | PTRMS | 15† counts/ppb | Aoki et al. (2007) |
| **$C_3$-$C_6$** | nitroperoxycarbonyl | $20^+$ | PTRMS | | D'Anna et al. (2005) |
| **Cycloalkanecarbaldehydes** | cycloalkyl nitrates | $H_3O^+(H_2O)_n$ | | counts/ppb | |

1  * Selected Ion Flow Drift Tube; † as specific alkyl ($-R^+$) fragment

2  PAN = peroxyacetyl nitrate; PPN =peroxypropionyl nitrate; PiBN = peroxyisobutyryl nitrate; MPAN = peroxymethacryloyl nitrate; APAN =

3  peroxyacryloyl nitrate; PnBN = peroxybutyryl nitrate; PBzN = peroxybenzoyl nitrate



*Table 2 Organic nitrates normalized sensitivity as a function of the ionization mode*

| Compound | MW (g mol⁻¹) | H₃O⁺ ionization | | | NO⁺ ionization | | |
|---|---|---|---|---|---|---|---|
| | | Signal (m/z) | Sensitivity (ncpm) ppbv⁻¹ | E/N (Td) | Signal (m/z) | Sensitivity (ncpm) ppbv⁻¹ | E/N (Td) |
| AlkC3 propyl nitrate | 105 | 43 | 35 | 70 | 104 | 1.1 | 34* |
| | | 59 | 18.8 | | 135 | 48.8 | |
| | | 106 | 2 | | | | |
| AlkiC4 isobutyl nitrate | 119 | 57 | 43.4 | 70 | 118 | 8.3 | 34* |
| | | 73 | 23 | | 149 | 78.9 | |
| | | 120 | 2 | | | | |
| KnC3 nitroxyacetone | 119 | 75 | 45.4 – 88.4 | 73 – 75 | 149 | 320 - 331 | 45* |
| | | 120 | 119 – 144 | 73 – 75 | | | |
| | | 137 | 127 | 48* | | | |
| KnC5 3-nitroxy-3-methyl-2-butanone | 147 | 103 | 38.3 | 74 | 177 | 249 - 265 | 39* |
| | | 148 | 8 | 74 | | | |
| | | 165 | 538 | 48* | | | |
| PAN | 121 | 122 | 7.7 | 90 | 151 | weak | |
| 1OH3C3 1-hydroxy-3-nitroxy-propane | 121 | 104 | 0.2 | 62 | 120 | 22.8 | 34* |
| | | 120 | 0.2 | | 151 | 268 | |
| | | 122 | 2 | | 167 | 10.2 | |
| | | 139 | 52.6 | | | | |
| | | 140 | 38 | | | | |

2  \* Without taking into account the RF contribution





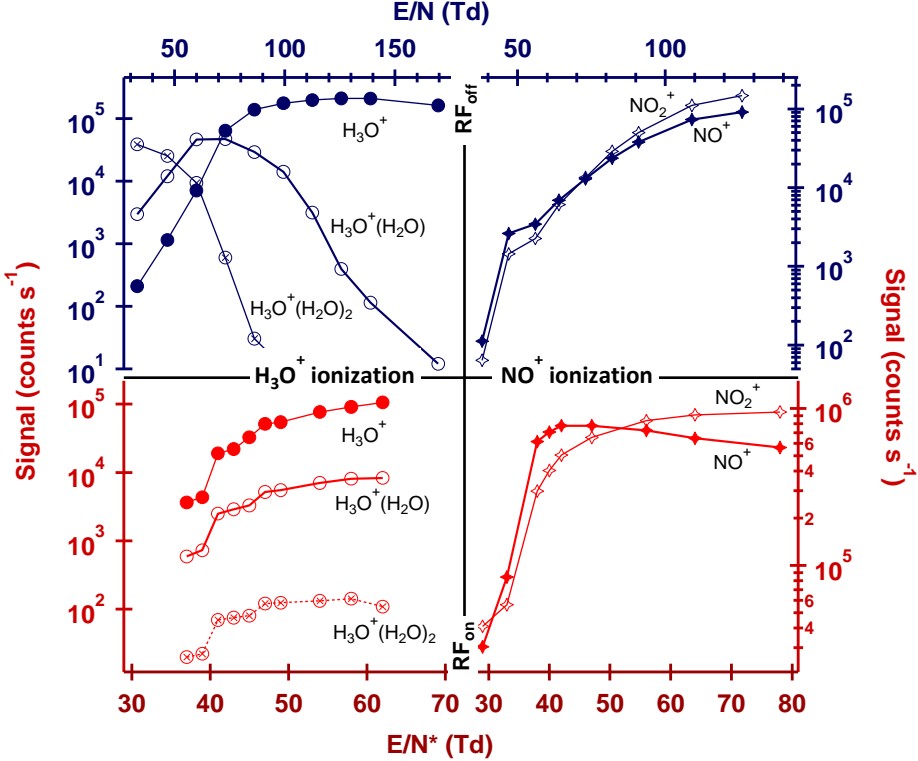

Figure 1. Typical ionizing species distribution as a function of the E/N ratio, in the two
ionization modes with RF mode on and off. When the RF mode is on, E/N* ratio were
calculated taking into account only the contribution of the dc electric field while the
additional input of the ac electric field remains difficult to estimate.



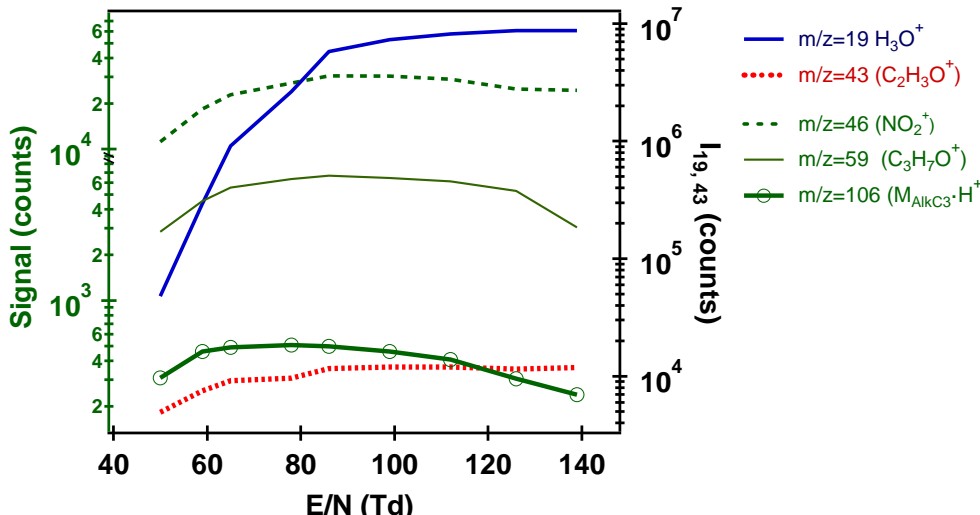

Figure 2. The E/N ratio influence over the AlkC3 identification for several representative
signals as recorded in absence of a RF funnel (left axis). The $H_3O^+$ ion is equally illustrated in
order to point the water cluster ions distribution influence, while the common signal at
m/z=43 is a fragmentation mark (right axis).





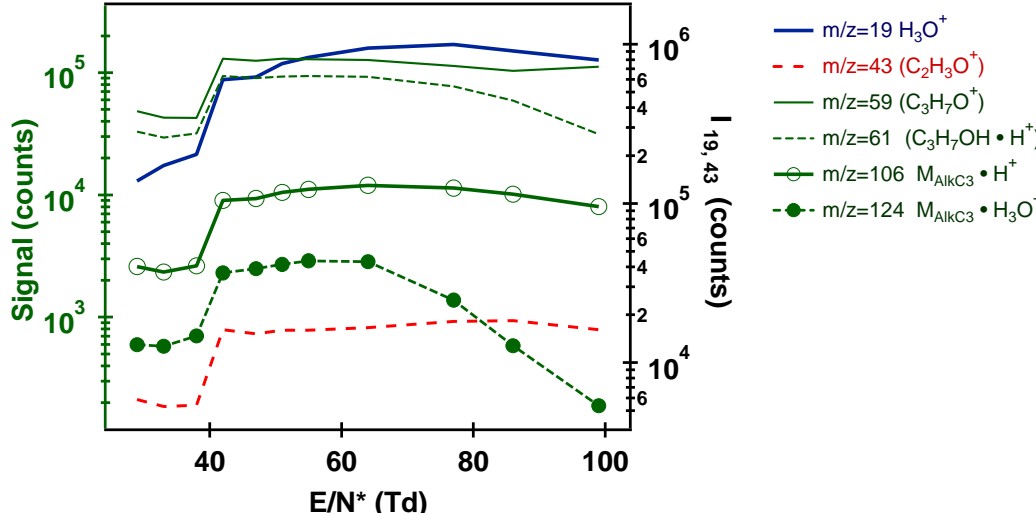

Figure 3. The E/N ratio influence over the AlkC3 identification for several representative
signals as recorded with the RF mode (left axis). The $H_3O^+$ ion is equally illustrated in order
to point the water cluster ions distribution influence, while the common signal at m/z=43 is a
fragmentation mark (right axis).



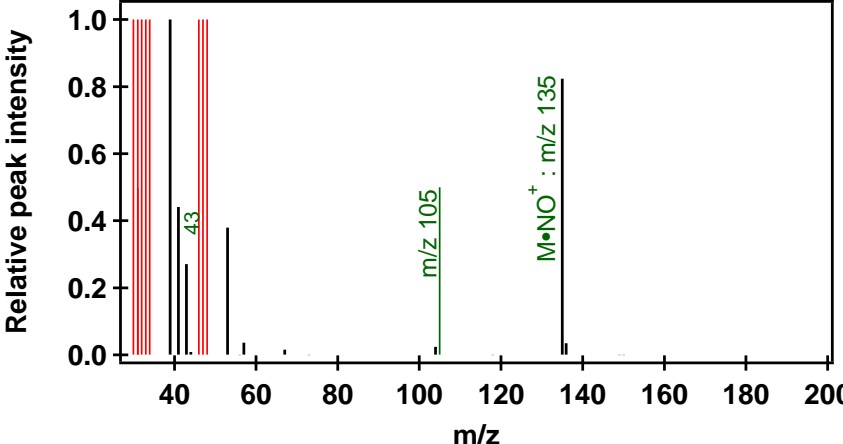

Figure 4. Recorded mass spectrum of AlkC3 (black bars) at the lowest extent of
fragmentation (E/N* = 34 Td) in the $NO^+$ ionization mode. The green thin line represents the
expected molecular ion of the analyte and the intense signals depicted by the red thin bars
represent the ionizing analytes at m/z = 30 ($NO^+$) and 46 ($NO_2^+$) and their isotopic abundance
signals at m/z = 31 and 47, 48 respectively.





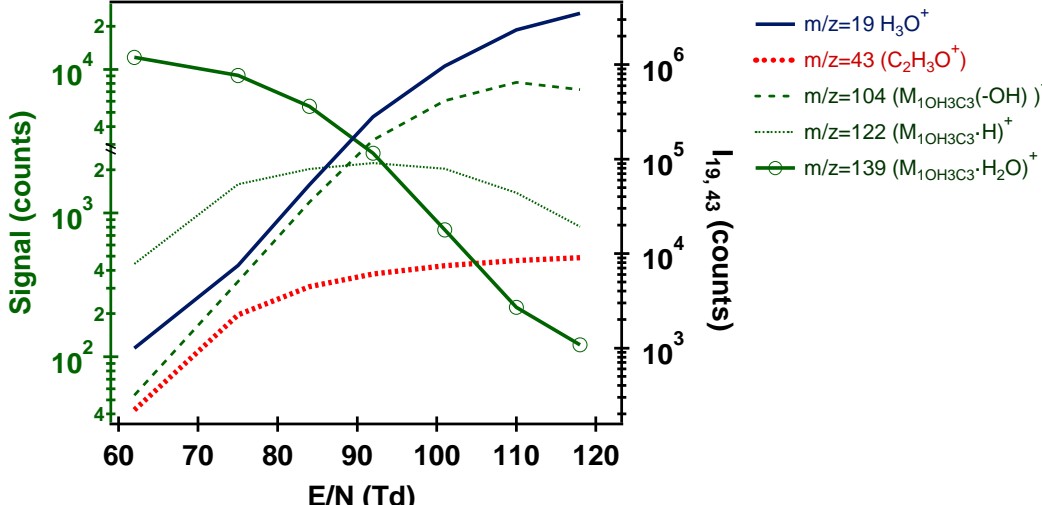

2  Figure 5. Typical signals due to the 1OH3C3 soft ionization and their behavior as a function
3  of the E/N ratio variation (left axis) in absence of a RF mode. The fragmentation intensity
4  depicted by the m/z=43 ion is illustrated together with the $H_3O^+$ ion onto the right axis.





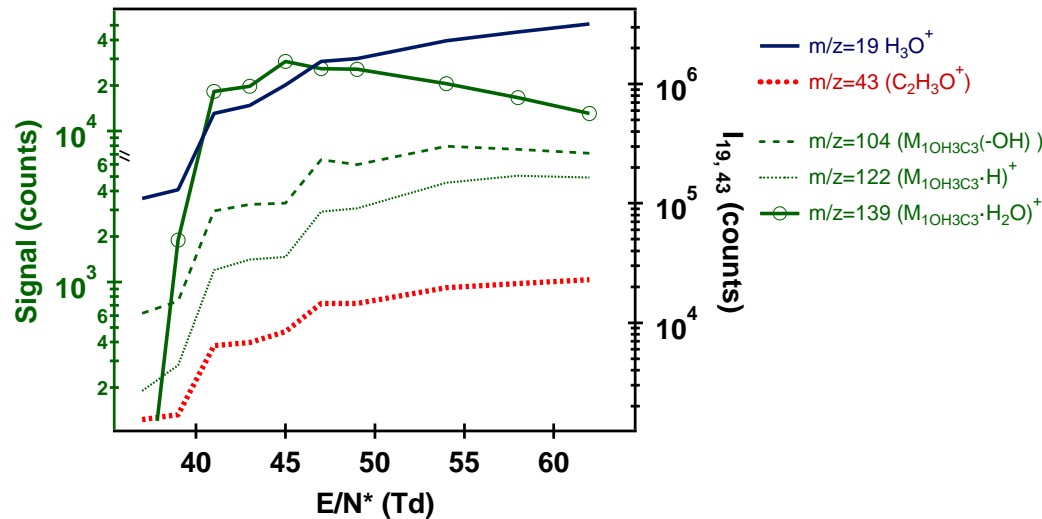

2  Figure 6. Typical signals due to the 1OH3C3 soft ionization and their behavior as a function
3  of the E/N ratio variation (left axis) in the RF mode. The fragmentation intensity depicted by
4  the m/z=43 ion is illustrated together with the $H_3O^+$ ion onto the right axis.



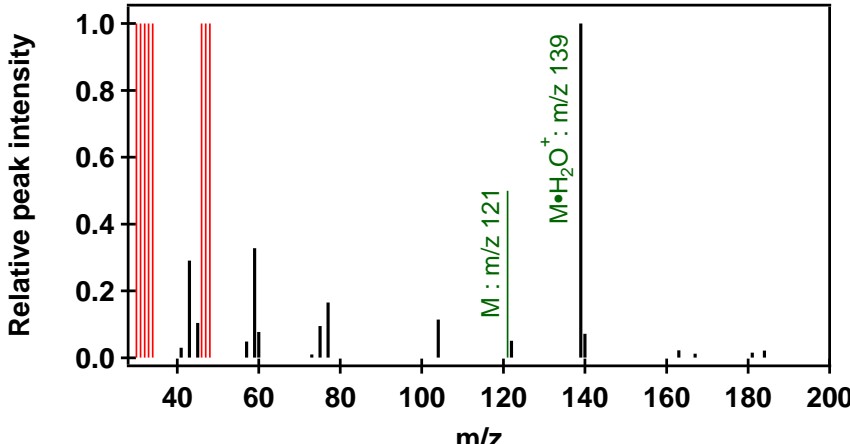

2 Figure 7. The recorded mass spectrum of 1OH3C3 (black bars) at the lowest extent of
3 fragmentation (E/N* = 45 Td) in the $H_3O^+$ ionization mode under the influence of the RF
4 mode.





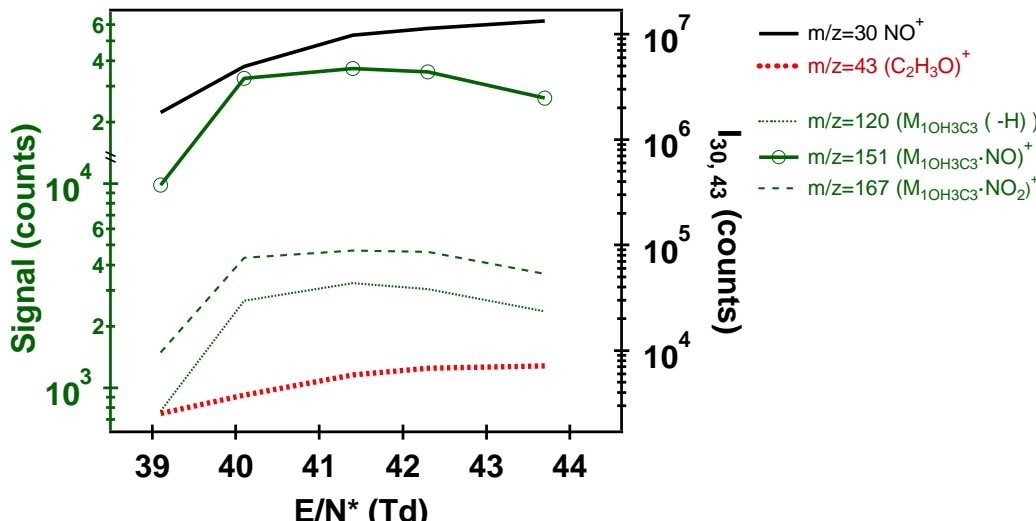

2  Figure 8. The E/N ratio and their influence over the 1OH3C3 ionization and identification
3  pathways, for several characteristic signals.



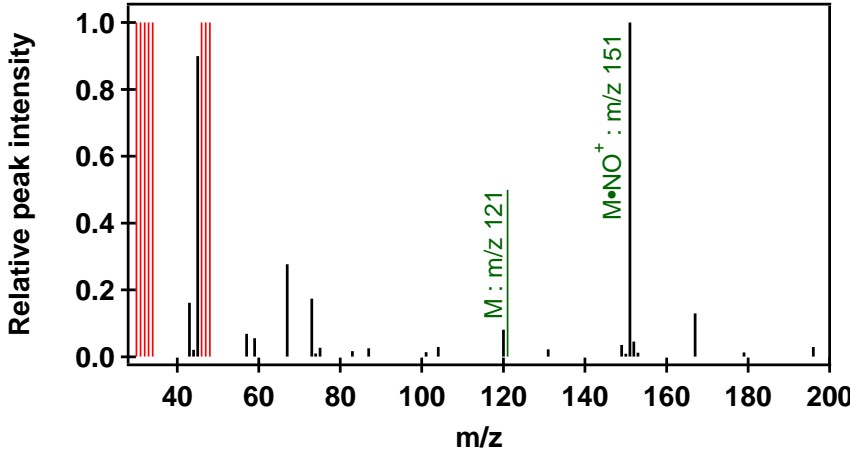

2 Figure 9. Recorded mass spectrum of 1OH3C3 (black bars) at the lowest extent of

3 fragmentation (E/N* = 41 Td) in NO⁺ ionization mode.



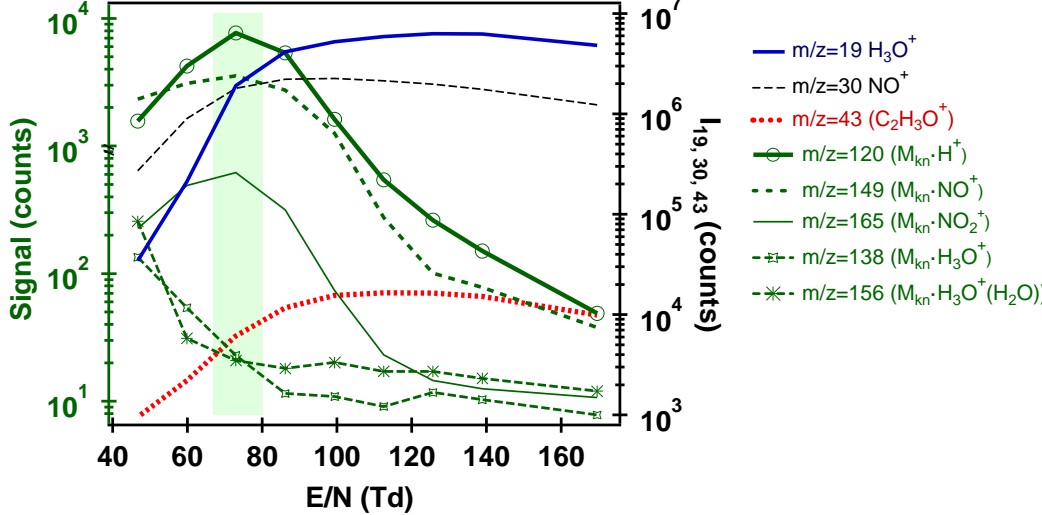

2 Figure 10. The KnC3 typical signals distribution over a wide range of E/N ratios (left axis),
3 illustrating the water cluster ions distribution impact over the keto-nitrates identification. The
4 fragmentation representative, m/z=43 signal, together with the $H_3O^+$ and $NO^+$ ionizing species
5 are equally plotted onto the right axis.





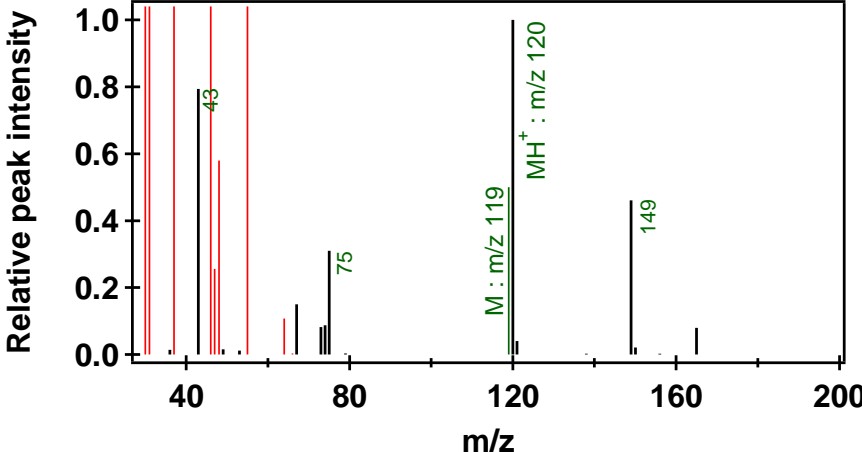

2    Figure 11. Recorded mass spectrum of protonated KnC3 (black bars) for E/N = 75 Td,
3    corresponding to the highest sensibility for the protonated analyte signal detection (m/z 120).



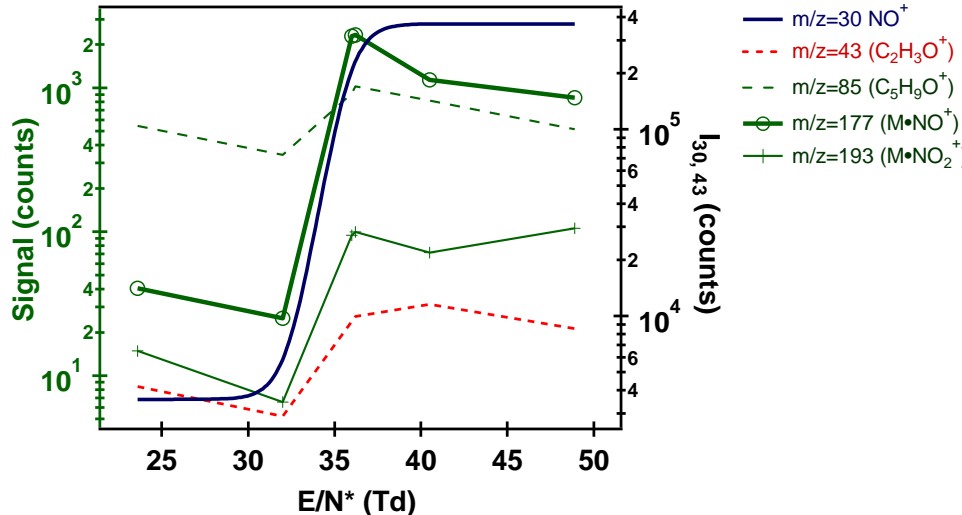

2 Figure 12. The RF mode behavior of typical $NO^+$ ionization signals of keto-nitrates (KnC5)
3 under the E/N ratio influence (left axis). Typical distribution of the $NO^+$ ions and $C_2H_3O^+$
4 fragment into the given E/N interval (right axis).





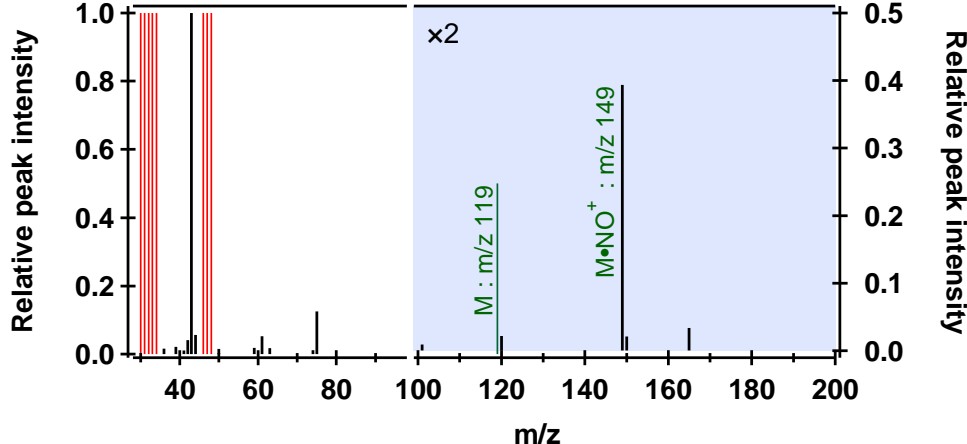

2    Figure 13. Recorded mass spectrum of KnC3 adduct (black bars) at the lowest extent of
3    fragmentation (E/N* = 40 Td) in the NO⁺ ionization mode.