# Peer review of "Figure S1. Recorded mass spectrum of AlkC3 (black bars) at the lowest extent of fragmentation of the molecular ion at $m/z = 106$ ( $E/N = 78$ Td) in the $\text{H}_3\text{O}^+$ ionization mode. The green thin line represents the expected molecular ion of the analyte and the intense signals depi"

_Atmospheric Measurement Techniques, 2016_

## Referee Comment (RC1) · Anonymous Referee #1 · 20 Dec 2016

**Review of Duncianu et al., 2016**

*General Comments:*
This paper presents interesting work on the application of proton transfer reaction mass spectrometry to the measurement of speciated organic nitrates using different compounds that represent different classes of organic nitrates (2 alkyl nitrates, 1 hydroxyl nitrate, 2 keto nitrates and PAN). The paper is well written, the data appears to be of good quality and the results are of scientific interest. I would support publication subject to corrections outlined in the comments below.

*Specific Comments:*
(1) On line 370 the purities of the synthesised keto-nitrates by FTIR should be given? Were these comparable to the commercially purchased alkyl nitrates? If not what were the major impurities and what are the likely effects on the results presented?

(2) On line 444, it is mentioned that the abundance of water clusters were higher than typical during this study. Is there a reason for this?

(3) In my opinion there are too many figures and the way they are shown prevents a more straightforward comparison of the different ionization schemes, RF modes and E/N ratios. The paper would be greatly improved if the results for each of the different types of organic nitrates (alkyl nitrates, hydroxyl nitrates and keto nitrates) were presented in a similar way to Figure 1 where operational differences can be clearly seen and compared. For example:

[Figure]

I note that the NO+ mode without RF (top right quadrant of this diagram) was less favourable to NO+ adduct formation and thus was not pursued in this work (lines 409-414). However, I would recommend including that data for completeness. With the same reasoning as above I would combined the mass spectra figures for the different ionization modes, e.g., for 10H3C3 figures 7 and 9. Again, if the data were available I would include the mass spectra for all modes and RFon/off, for the optimised E/N ratio in each case to give the complete picture.

(4) The optimised conditions appear to be slightly different for each class of organic nitrates and this clearly limits this technique in terms of which compounds can be observed simultaneously as I believe this is a parameter that is not possible to change 'on the fly' in contrast to the reagent ion that can be switched as in the switchable reagent ion (SIR) versions of the PTRMS instrument. Table 2 summaries the optimisation work and shows the sensitivities. However, I think there is a final step missing here. To be the most useful analytical tool it is better to encompass the widest range of analytes, in this case all of the organic nitrates. Thus it seems that what should also be included is a recommendation for

operation of the PTRMS to best detect all the different types of organic nitrates discussed here. What are these conditions, i.e., NO+ with E/N of ~40, what are the sensitivities of the different types (alkyl nitrates, hydroxyl nitrates, keto nitrates and PANs) under this single set of analytical conditions?

(5) In the introduction there is discussion of the limited measurements of speciated organic nitrates (especially multifunctional ones) and thus some of the motivation for developing this approach is to provide a method to measure these compounds in the atmosphere. However, data is only presented from a smog chamber containing the specific target compounds and not a more complex mixture of compounds that would present an analytical challenge. However, to really convince the reader of the application of this approach to ambient measurements there really needs to be some actual ambient measurement data included in the manuscript. This would really strengthen the paper and I strongly encourage the authors to do it. If not then at the very least there needs to be a very convincing discussion with reference to published ambient measurement data to demonstrate how unique the mass fragments/molecular ions are relative to the mass fragments/molecular ions of other constituents of the ambient atmosphere that are also ionized within this instrument. For example, line 471 (H3O+ ionization of AlkC3) mentions RO+ m/z 59, in the atmosphere this signal is likely dominated by acetone. Without this it is difficult to assess the applicability of this approach to ambient measurements.

(6) The data presented only refers to unit mass resolution data, was the high accuracy mass data also used? I could imagine that it might be useful in distinguishing the organic nitrates from other atmospheric compounds that are also ionised by the PTRMS.

(7) PAN type compounds are sensitive to thermal decomposition. With the inlet at 40 degrees C one might expect some thermal decomposition of PANs (dependent on residence time in the inlet)? Can the authors comment on this and whether this may be a reason for the high detection limits observed for these compounds with this method.

***Technical Corrections:***
Line 25 – reword '…allows to easily identify the organic nitrate (R) with the….' to read '…allows the easy identification of the organic nitrate (R) from the….'.

Line 30 – reword 'This method exhibits however lower capabilities for the detection…..' to read 'However, this method exhibits much lower capabilities for detection….'.

Line 34 – reword 'Organic nitrates are important….' to read 'Organic nitrates are an important….'.

Line 35 and throughout the manuscript – subscript of x in the abbreviation NOx

Line 39 – reword 'They play therefore…' to read 'They therefore play….'.

Line 41 – reword '….impact of organic nitrates chemistry on ozone budget….' to read 'impact of organic nitrate chemistry on the ozone budget….'.

Line 44 and throughout the manuscript - subscript of y in the abbreviation NOy

Line 46 – reword 'They are monofunctional alkyl nitrates, PANs but also….' to read 'They are not only monofunctional alkyl nitrates and PANs but also…..'.

Line 48 – reword 'These latter include i) hydroxynitrates which are formed by the oxidation of alkenes initiated by OH radicals and by isomerisation processes of alkoxy radicals….' to read 'The

latter include i) hydroxynitrates formed by the OH oxidation of alkenes followed by isomerisation processes of the resultant alkoxy radicals….'.

Line 51 – reword '…alkenes but are also second-generation….' to read '…alkenes and also as second-generation….'.

Line 57 – remove 'then'

Line 58 – replace 'commercial' with 'commercially available'.

Lines 71 to 75 – these lines are confusing as written. Suggest rewording 'The thermal dissociation (TD) properties of different classes of nitrates was used as an analytical tool to sketch the global chemistry of $RONO_2$, same as the Laser-Induced Florescence (LIF), cavity ring down spectroscopy (CRDS)(Paul et al., 2009), or cavity attenuated phase shift (CAPS)(Sadanaga et al., 2016), were used to quantify the $NO_2$ issued from the organic nitrates decomposition.' to read something like 'The thermal dissociation (TD) properties of different classes of nitrates was used as an analytical tool to probe the global chemistry of $RONO_2$ utilizing Laser-Induced Florescence (LIF), cavity ring down spectroscopy (CRDS)(Paul et al., 2009), or cavity attenuated phase shift (CAPS)(Sadanaga et al., 2016) to quantify the $NO_2$ evolved from organic nitrate thermal decomposition'.

Line 151 – replace 'Worth noticing…' with 'It is worth noting…'

Line 178 (also Line 638) – replace 'Worth notice…' with 'It is worth noting…'

Line 273 – replace 'thoughtful' with 'well thought out' or 'carefully thought out'.

Line 312 – LISA, define abbreviation on first use.

Line 377 – replace 'leads' with 'leading'.

Line 522 – replace '..used as template…' with '…used as an example…' or '…used as a reference….'

Line 742 – Missing citation, there appears instead an error (Erreur! Source du renvoi introuvable)

Line 778 and throughout manuscript – use of the word 'sensibility' should be replaced with 'sensitivity'. (e.g., lines 665, 667,

Line 780 – reword '…signals corresponding to the adduct formation is barely noticed….' to read as '…signals corresponding to adduct formation are barely noticeable….'.

Line 807 – replace '…deployment of the…' with '…using this…'

---

## Referee Comment (RC2) · Anonymous Referee #2 · 2 Jan 2017

The paper "Measurement of alkyl and multifunctional organic nitrates by Proton Transfer Reaction Mass Spectrometry" by Duncianu et al., investigates the organic nitrates in the gas phase using an optimized PTR-ToF-MS instrument at tens of ppt detection limit range. The paper is well written with high scientific quality data and deserves publication after considering the revisions.

Comments: 1. The authors mentioned the synthesis of few organic nitrates performed during these investigations. However, the purity of the synthesized compounds is verified only by FTIR as the authors stated in line 370. Please consider to add in Supporting information section the IR gas phase spectra of the synthesized compounds. Additionally, please consider that FTIR analysis is a less appropriate method for checking

purity of the new compounds when the impurities may have similar functional groups as the synthesized compound. Did you perform NMR analysis of the synthesized compounds? If you compared the synthesized compounds with existing IR reference spectra please specify the reference of the database used. Additionally, please add information of the infrared gas phase cross section values for the organic nitrates obtained in this study as you mentioned to calculate them (line 339-351).

2. There is stated in the Abstract and Introduction sections that was performed optimization of the PTR-ToF-MS instrument for the measurements of organic nitrates in the atmosphere but the studies presented in the manuscript are exclusively recorded in a simulation chamber using dry air conditions. My concern related to the measurements into the atmosphere is mainly due to the interference species (gas and particles) and humidity. Did you perform any test for humidity effect? May you comment how these interference could affect the results (decomposition, clustering, partition, hydrolysis, etc. )?

3. As there is a FTIR instrument available for insitu measurements in the chamber, it will be worth comparing FTIR vs PTR-MS concentration-time dependency for each class of organic nitrates at the most effective PTR-MS conditions during the organic nitrate accommodation into the chamber. These comparisons may add important information for quantitative analysis of organic nitrates using PTR-MS. Such test may also check for possible response delay in the PTR measurements due to the nitrates deposition on the sampling line for example.

4. In the Conclusion section, authors conclude that NO+ adduct ionization mode is "suitable for measurement of organic nitrates during lab studies but also in ambient air". There are tests performed in the ambient air?

Specific and technical:

Line 34: Please use subscript for NOy and NOx in entire manuscript.

Line 51 Please add a reference for the formation of carbonyl-nitrates by NO3-oxidation of alkenes.

Line 174: The AC and DC electrical field better symbolize with capital letters and explain the ac and dc meaning.

Line 183: Provide more details for the sampling line (diameter, etc). Heating up to 40° C prevents adsorption for all investigated species? Have you performed a sampling efficiency study for different compounds (monofunctional organic nitrates, hydroxyl nitrates, dinitrates, carbonyl nitrates, etc)? Please comment on this sampling efficiency.

Line 188: The PTR-MS inlet system has been optimized for various temperatures to study possible decomposition processes of the organic nitrates? If yes, there is a need for some more comments.

Line 312: please express the meaning of LISA abbreviation.

Line 338: add "." after "al" in "Doussin et al, 1997"

Line 343: please revise the units for IBIs. There is "cm molecule-1"

Line 347: Beaver et al., 2012 and Bates et al., 2014 do not provide IBIs values for the PANs and hydroxynitrates. Please revise.

Line 361: "H2O < 0,5 ppm" use dot instead comma

Line 362: "CO < 0,1 ppm" use dot instead comma

Line 742: there is a text which should be erased.

Line 752: "C5H7-C(O)OONO2" is in fact "C4H7-C(O)OONO2"

Reference: - Please be consistent with journal title and use abbreviated or full name but not mixed. - Please be consistent with using capital letter or not for the words forming the title of the articles. - Please move "Shepson" from end of reference list to the right place.

---

## Author Comment (AC1) · 31 Jan 2017

**General comments to the referees**

First of all, thank you very much to the anonymous referees for this interactive discussion and their productive comments, corrections and suggestions that ensued. Here we have carefully replied to all comments and the paper has been improved following the recommendations of reviewers.

**(A) Comments from Referees**

**Ref1. C1.** On line 370 the purities of the synthetized keto-nitrates by FTIR should be given? Were these comparable to the commercially purchased alkyl nitrates? If not what were the major impurities and what are the likely effects on the results presented

**Ref2. C1.** The authors mentioned the synthesis of few organic nitrates performed during these investigations. However, the purity of the synthesized compounds is verified only by FTIR as the authors stated in line 370. Please consider to add in Supporting information section the IR gas phase spectra of the synthesized compounds. Additionally, please consider that FTIR analysis is a less appropriate method for checking purity of the new compounds when the impurities may have similar functional groups as the synthesized compound. Did you perform NMR analysis of the synthesized compounds? If you compared the synthesized compounds with existing IR reference spectra please specify the reference of the database used. Additionally, please add information of the infrared gas phase cross section values for the organic nitrates obtained in this study as you mentioned to calculate them (line 339-351).

**(B) Author's response**

Once the functionalized compounds have been synthesized, a recurrent head-space purification step was performed in order to extract the lower volatility byproducts of the synthesis (line 367). The reactor injections were subsequently performed using a vacuum line allowing the quantification of a known volume of analyte before injection. The procedure allows the graduate "distillation" of the bulk synthesis and the injection of the useful fraction in the simulation chamber. Complementary analysis showed that subsequent injected analyte by subsequent headspace sample are gradually increasing the purity level of the injected analyte by subsequent headspace samplings. Therefore is difficult for us to estimate the bulk purity of the synthesis. In addition, it was checked by GC-MS that the reactant used for the synthesis of the organic nitrates (eg. an hydroxy-ketone for the synthesis of a keto-nitrate) was fully consumed preventing any interference in mass spectra.

The remaining main by-product present in the same distillation fraction as the organic nitrate is HNO3 representing up to 5% of the injected volume. Other impurities can be observed such as the HCOOH, NO2 or the HONO in the recorded IRTF spectra, representing less than 1% of the injected volume. All these impurities are readily identified and subtracted from the IRTF spectra in order to correctly quantify the organic nitrate absorbance. The impurities may however add supplementary signals in the PTRMS mass spectra. These signals are systematically at lower masses than the ones regarding the main ionization processes of the organic nitrates and therefore won't interfere. However we cannot exclude that some of the PTRMS signals assigned as organic nitrate fragments could have a more or less important contribution from these impurities. Efforts have been made in order to quantify and identify the mass spectrum double peaks into the same mass unit which can be due of this kind of interference.

The FTIR gas-phase spectra of the synthesized organic nitrates were added in supplementary information section and compared with a commercially available alkyl nitrate spectrum. The NMR analysis was not available during the project.

The infrared cross sections for the organic nitrates obtained in this study are mentioned (line 343-348) for the absorption band centered at 820 cm-1 as being the less subject to interferences.

**(C) Author's changes in manuscript.**

*Line 369 added:* In addition, it was checked by GC-MS that the organic reactant used for the synthesis, here the hydroxyl-ketone, was fully consumed preventing any interference in IR and mass spectra.

*Line 377 added:* The total consumption of the bromo-alcohol was checked by GC-MS.

**Paragraph to be added in the Chemicals and gases:**

The presence of impurities (mainly  $HNO_3$  and HCOOH) as synthesis by-products may add supplementary signals in the PTRMS mass spectra. These signals are however systematically at lower masses than the ones regarding the main ionization processes of the organic nitrates and therefore won't interfere in the main frame of the discussion.

Figure to be added in SI

Figure S1. The FTIR gas phase absorbance spectra of the synthesized organic nitrates (KnC3, KnC5, 10H3C3) compared with a commercially available alkyl nitrate spectra (AlkC3).

**Ref1. C2.** On line 444 it is mentioned that the abundance of the water clusters were higher than typical during this study. Is there a reason for this?

**(B) Author's response**

As stated in lines 395-402, and depicted in Figure 1, using a PTRMS instrument at atypical E/N ratios, inferior to 100Td, will rather favor the water clusters formation than the hydronium ions. This condition seems more favorable to the organic nitrate ionization without complete fragmentation.

**(C) Author's changes in manuscript.**

Line 443 reword "..." to read "At higher abundances of the water clusters, expected to occur in our study due to the usage of low E/N ratios, the R5 mechanism is considered..."

**Ref1. C3.** In my opinion there are too many figures and the way they are shown prevents a more straightforward comparison of the different ionization schemes. The paper would be greatly improves if the results for each of the different types of organic nitrates were presented in a similar way to Figure 1 were operational differences can be clearly seen and compared.

**(B) Author's response**

Indeed the suggested changes will significantly improve the appearance of the paper and summerize the obtained results. As you already noticed the top right quadrant of the diagram will not be plotted since the lack of signals in this operational condition forced us to abandon the recordings. The data isn't available for these conditions.

**(C) Author's changes in manuscript.**

*Figures 2,3 and supplementary data are condensed and displayed in the new plot regarding alkyl nitrate ionization modes*

---

## Author Comment (AC2) · 31 Jan 2017

Please find attached the authors answers to both referees comments.

Please also note the supplement to this comment:
http://www.atmos-meas-tech-discuss.net/amt-2016-318/amt-2016-318-AC2-supplement.pdf